# Estimation of Stochastic Optimal Transport Maps

**Sloan Nietert**
EPFL
sloan.nietert@epfl.ch

**Ziv Goldfeld**
Cornell University
goldfeld@cornell.edu

## Abstract

The optimal transport (OT) map is a geometry-driven transformation between high-dimensional probability distributions which underpins a wide range of tasks in statistics, applied probability, and machine learning. However, existing statistical theory for OT map estimation is quite restricted, hinging on Brenier's theorem (quadratic cost, absolutely continuous source) to guarantee existence and uniqueness of a deterministic OT map, on which various additional regularity assumptions are imposed to obtain quantitative error bounds. In many real-world problems these conditions fail or cannot be certified, in which case optimal transportation is possible only via stochastic maps that can split mass. To broaden the scope of map estimation theory to such settings, this work introduces a novel metric for evaluating the transportation quality of stochastic maps. Under this metric, we develop computationally efficient map estimators with near-optimal finite-sample risk bounds, subject to easy-to-verify minimal assumptions. Our analysis further accommodates common forms of adversarial sample contamination, yielding estimators with robust estimation guarantees. Empirical experiments are provided which validate our theory and demonstrate the utility of the proposed framework in settings where existing theory fails. These contributions constitute the first general-purpose theory for map estimation, compatible with a wide spectrum of real-world applications where optimal transport may be intrinsically stochastic.

## 1   Introduction

Optimal transport (OT) is a principled framework for comparing and transforming probability distributions according to the geometry of the underlying metric space [Villani, 2003, Santambrogio, 2015]. Central to OT theory is the transport map, which performs said transformation. For $\mathcal{X}, \mathcal{Y} \subseteq \mathbb{R}^d$, we say that $T : \mathcal{X} \to \mathcal{Y}$ is a *transport map* from source distribution $\mu \in \mathcal{P}(\mathcal{X})$ to target $\nu \in \mathcal{P}(\mathcal{Y})$ if the pushforward measure $T_\sharp \mu = \mu \circ T^{-1}$ coincides with $\nu$. An optimal transport map $T^\star$, if it exists, is a solution to the *Monge OT problem* from $\mu$ to $\nu$, which reads as follows for the $p$-Wasserstein cost:

$$\inf_{T : T_\sharp \mu = \nu} \mathbb{E}_\mu[\|X - T(X)\|^p]^{\frac{1}{p}}. \tag{1}$$

Monge maps are employed for many applications, including domain adaptation [Courty et al., 2017, Redko et al., 2019], single-cell genomics [Schiebinger et al., 2019, Bunne et al., 2023], style transfer [Kolkin et al., 2019, Mroueh, 2020], and generative modeling [Zhang et al., 2018, Vesseron et al., 2025]. An important special case is when $p = 2$ and $\mu$ is absolutely continuous with respect to (w.r.t.) the Lebesgue measure; then, Brenier's theorem guarantees the existence of a unique Monge map, often called the *Brenier map*, given as the gradient of a convex potential [Brenier, 1991]. More generally, existence of optimal maps is guaranteed if $\mu$ is absolutely continuous and uniqueness holds if further $p > 1$ (see, e.g., Section 2.4 of Villani, 2003).

There is a rich literature on formal guarantees for estimation of Brenier maps [Hütter and Rigollet, 2021, Pooladian and Niles-Weed, 2021, Deb et al., 2021, Manole et al., 2024] (see related work).

39th Conference on Neural Information Processing Systems (NeurIPS 2025).

However, all of these works impose stringent regularity assumptions on the density of $\mu$ (e.g., two-sided bounds) and/or the unique Brenier map $T^\star$ (e.g., Lipschitzness and Hölder smoothness). This is because the quality of the estimator is measured by its $L^p(\mu)$ distance from $T^\star$, which inherently requires uniqueness (otherwise, the $L^p(\mu)$ metric is meaningless) and hinges on said regularity assumptions to obtain quantitative error bounds. However, such regularity assumptions are often impossible to verify in practice. Worse yet, many real-world applications violate the conditions of Brenier's theorem, whence deterministic Monge maps may not exist, and optimal transportation strategies require stochasticity. For instance, this is the case in domain adaptation whenever the source distribution lies on a lower-dimensional manifold than the target [Courty et al., 2017, Redko et al., 2019], such as in text-to-image or sketch-to-photo translation. Similarly, single-cell developmental trajectories branch over time, so any measure-preserving map from an early snapshot to a later one must be stochastic [Schiebinger et al., 2019]. As such scenarios far exceed the account of current OT map estimation theory, this work sets out to close this gap by providing a broadly applicable estimation framework that offers strong recovery guarantees for a breadth of applications.

## 1.1 New Framework for Stochastic OT Map Estimation and Contributions

The Kantorovich OT problem [Kantorovich, 1942] relaxes that of Monge by allowing stochastic maps. Reparametrizing the standard formulation via couplings in terms of Markov kernels, it reads as

$$
\mathsf{W}_p(\mu, \nu) = \min_{\substack{\kappa \in \mathcal{K}(\mathcal{X}, \mathcal{Y}) \\ \kappa_\sharp \mu = \nu}} \left( \iint \|x - y\|^p \mathrm{d}\kappa(y|x) \mathrm{d}\mu(x) \right)^{\frac{1}{p}}, \quad (2)
$$

where $\kappa(\cdot|\cdot)$ varies over Markov kernels (regular conditional probability distributions) from $\mathcal{X}$ to $\mathcal{Y}$ and $\kappa_\sharp \mu$ denotes the pushforward measure $\int \kappa(\cdot|x)\mathrm{d}\mu(x)$.[1] We propose a novel framework for stochastic OT map estimation by furnishing a suitable error metric. For source distribution $\mu \in \mathcal{P}(\mathcal{X})$, target distribution $\nu \in \mathcal{P}(\mathcal{Y})$, and kernel $\kappa$ from $\mathcal{X}$ to $\mathcal{Y}$, we define the *transportation error* $\mathcal{E}_p(\kappa; \mu, \nu)$ of $\kappa$ for the $\mathsf{W}_p(\mu, \nu)$ problem by

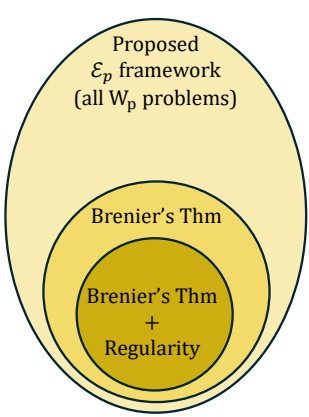

Proposed
$\mathcal{E}_p$ framework
(all $\mathsf{W}_p$ problems)

Brenier's Thm

Brenier's Thm
+
Regularity

$$
\underbrace{\left[ \left( \iint \|x - y\|^p \mathrm{d}\kappa_x(y)\mathrm{d}\mu(x) \right)^{\frac{1}{p}} - \mathsf{W}_p(\mu, \nu) \right]_+}_{\text{optimality gap}} + \underbrace{\mathsf{W}_p(\kappa_\sharp \mu, \nu)}_{\text{feasibility gap}},
$$

where $[c]_+ := \max\{c, 0\}$ and $\kappa_x(\cdot) := \kappa(\cdot|x)$. Under $\mathcal{E}_p$, the quality of $\kappa$ is thus measured by its transportation cost overhead on top of the optimum $\mathsf{W}_p(\mu, \nu)$ (dubbed *optimality gap*), plus its $p$-Wasserstein gap from matching the target $\nu$ (the *feasibility gap*). While the $\mathcal{E}_p$ error metric naturally accounts for deterministic OT maps, it does not require uniqueness or even existence thereof.

Figure 1: Previous map estimation theory only accounts for the inner circle, despite many OT map applications lying outside it. The proposed estimation framework under $\mathcal{E}_p$ covers all possible $\mathsf{W}_p$ problems (subject to tail bounds for quantitative rates).

This enables treating OT map estimation settings far beyond those accounted by existing theory, as illustrated in Figure 1 to the right. Remarkably, beyond the broad coverage of the proposed framework, quantitative bounds on $\mathcal{E}_p$ can be derived under minimal and easy-to-verify assumptions, rendering the guarantees applicable in practice.

Our technical contributions build upon a foundation of stability lemmas for $\mathcal{E}_p$ established in Section 2. These characterize how $\mathcal{E}_p$ responds to TV and Wasserstein perturbations of the input measures and to compositions of the kernel. In Section 3, we apply these to finite-sample estimation and computation. Here, our strongest result holds when $\nu$ is sub-Gaussian and $\mu$ has bounded $2p$th moments, but assuming no regularity of an optimal kernel. For i.i.d. samples $X_1, \ldots, X_n \sim \mu$ and $Y_1, \ldots, Y_n \sim \nu$, we present a rounding-based estimator $\hat{\kappa}_n$ which achieves $\mathbb{E}[\mathcal{E}_p(\hat{\kappa}_n; \mu, \nu)] = \widetilde{O}_{p,d}\big(n^{-1/(d+2p)}\big)$, with running time $O(n^{2+o_d(1)})$ dominated by a single, low-accuracy call to an entropic OT solver. We also observe a minimax lower bound of $\Omega(n^{-1/(d \vee 2p)})$, showing that our rate is near-optimal.

---

[1] The standard Kantorovich OT problem optimizes the cost over couplings $\pi \in \Pi(\mu, \nu)$, but the disintegration theorem yields that each such coupling can be decomposed as $\mathrm{d}\pi(x, y) = \mathrm{d}\mu(x)\mathrm{d}\pi(y|x)$, where $\pi(\cdot|\cdot)$ is a Markov kernel induced by conditioning on the left argument. When a coupling is induced by a deterministic map $T$, i.e., $\pi = (\mathrm{Id}, T)_\sharp \mu$, the corresponding kernel $\kappa$ is given by $\kappa_x = \delta_{T(x)}$.

In Section 4, we examine the statistical landscape of estimation when there exists a Hölder continuous optimal kernel, a condition that is still significantly weaker than typical assumptions for Brenier map estimation when $p = 2$. In particular, for the case of a Lipschitz optimal kernel $\kappa_\star$, i.e., when $\mathsf{W}_p((\kappa_\star)_x, (\kappa_\star)_{x'}) \lesssim \|x - x'\|$ for all $x, x' \in \mathcal{X}$, we show that kernel estimation under $\mathcal{E}_p$ has the same statistical complexity as estimating $\mu$ and $\nu$ under $\mathsf{W}_p$, with rate $\widetilde{O}(n^{-1/(d\vee 2p)})$. For this reduction, we employ an estimator based on Wasserstein distributionally robust optimization.

In Section 5, we show that effective kernel estimation is possible in the presence of adversarial data contamination. Historically, robust statistics has been well-studied under Huber's $\varepsilon$-contamination model for global outliers, which is subsumed by TV $\varepsilon$-corruptions of the input data [Huber, 1964]. More recently, statisticians have examined robust estimation under localized Wasserstein corruptions of the input samples [Zhu et al., 2022, Chao and Dobriban, 2023, Liu and Loh, 2023]. We consider a strong corruption model where the clean samples can be corrupted both in TV and under the Wasserstein metric. This combination of local and global corruptions only has only explored recently [Nietert et al., 2023b, 2024, Pittas and Pensia, 2024], and their interaction has required careful analysis. Here, stability of $\mathcal{E}_p$ enables us to cleanly decouple the two corruption types. Against an adversary with TV budget $\varepsilon$ and $\mathsf{W}_p$ budget $\rho$, we show that a convolutional estimator achieves error $\sqrt{d}\varepsilon^{1/p} + \sqrt{d}\rho^{1/(p+1)} + O_{p,d}(n^{-1/(d+2p)})$. An accompanying minimax lower bound of $\sqrt{d}\varepsilon^{1/p} + d^{1/4}\rho^{1/2} + n^{-1/(d\vee 2p)}$ implies a separation between robust map estimation under $\mathcal{E}_p$ and robust distribution estimation under $\mathsf{W}_p$, where one can achieve linear dependence on $\rho$.

In Section 6 , we validate our theory with numerical simulations for two settings with irregular OT maps that are poorly suited for existing theory. These showcase the performance of our rounding estimator and the benefits of $\mathcal{E}_p$ over $L^p$. Overall, our results constitute a general-purpose theory for (possibly stochastic) OT map estimation, subject to minimal primitive assumptions. As such, it is capable of providing formal performance guarantees in the $\mathcal{E}_p$ sense in various practically relevant settings.

**Related work.** Most related to this paper is a line of statistics work on the minimax sample complexity of Brenier map estimation when $p = 2$, initiated by Hütter and Rigollet [2021]. Under density assumptions on $\mu$ and smoothness conditions on the unique Brenier map $T^\star$ (in particular, Lipschitzness), they obtain near-optimal risk bounds of the form $\|\hat{T} - T^\star\|_{L^2(\mu)} = \widetilde{O}(n^{-1/d})$, using empirical risk minimization for a semi-dual objective. The myriad of follow-ups include Pooladian and Niles-Weed [2021], Deb et al. [2021], Manole et al. [2024], all of which impose density and smoothness assumptions. Pooladian et al. [2023] considered the semi-discrete setting where the Brenier map is piecewise constant, employing an estimator based on entropic OT (EOT). Recently, Balakrishnan and Manole [2025] provided refined guarantees that sidestep the typical density assumptions, but they still rely on the Brenier map being the gradient of a sufficiently regular convex potential. Lastly, a variety of neural map estimators have been developed by the machine learning community [Seguy et al., 2018, Meng et al., 2019, Wang and Goldfeld, 2024], with applications to domain adaptation, style transfer, trajectory estimation, and the like.

Two recent approaches for neural map estimation warrant further discussion. First is the Monge gap regularizer of Uscidda and Cuturi [2023]. For the $p$-Wasserstein cost, this work proposes training a deterministic map estimator to minimize the objective $\mathcal{J}_p(T; \mu, \nu) = \mathcal{M}_p(T; \mu) + \mathsf{D}(T_\sharp \mu, \nu)$, where D is a statistical divergence and the *Monge gap* $\mathcal{M}_p$ is defined by

$$\mathcal{M}_p(T; \mu) := \int \|x - T(x)\|^p \, \mathrm{d}\mu(x) - \mathsf{W}_p(\mu, T_\sharp \mu)^p. \tag{3}$$

They show that $\mathcal{M}_p \geq 0$ with equality if and only if $T$ is $c$-cyclically monotone over $\mathrm{supp}(\mu)$. Consequently, $\mathcal{J}_p$ nullifies exactly when $T$ is optimal for the $\mathsf{W}_p(\mu, \nu)$ problem. The statistical analysis of that work accounts for consistency, under the assumption that a deterministic and continuous optimal map exists. In practice, they suggest taking D as an EOT cost, estimating $\mathsf{W}_p$ with EOT, and substituting $\mu$ and $\nu$ with their empirical measures. Parameterizing $T$ via a multilayer perceptron, they achieve competitive empirical performance on a range of map estimation tasks. As we will show in Section 2, $\mathcal{E}_p$ and $\mathcal{J}_p$ are very connected; in particular, they coincide up to constant factors when $p = 1$ and $\Delta = \mathsf{W}_1$. We view $\mathcal{E}_p$ as better suited for quantitative statistical analysis, enabling rates which seem difficult to prove under $\mathcal{J}_p$ for general $p$ (and we are unaware of any existing rates proven under $\mathcal{J}_p$). On the other hand, as discussed in Section 6, we find that $\mathcal{J}_p$ is better suited for neural implementation, since its gradients seem to carry a stronger signal when far from optimality.

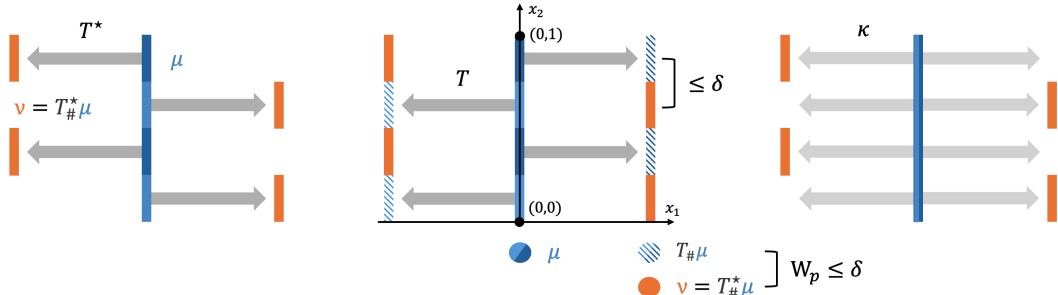

Figure 2: Diagrams of 2 maps and a kernel for $\mathsf{W}_p(\mu, \nu)$, where in each $\mu$ is uniform over the blue line connecting $(0, 0)$ and $(0, 1)$ and, in orange, $\nu = T^\star_\sharp \mu$ for $T^\star(x) = ((-1)^{\lfloor x_2/\delta \rfloor}, x_2)$. We depict $T^\star$ on the left, $T(x) = (-(-1)^{\lfloor x_2/\delta \rfloor}, x_2)$ in the center, and kernel $\kappa_x = \mathrm{Unif}(\{(-1, x_2), (1, x_2)\})$ on the right. While $T$ and $\kappa$ are far from $T^\star$ in an $L^p$ sense, they achieve $\mathcal{E}_p \leq \delta$ (indeed, both $T$ and $\kappa$ achieve zero optimality gap and at most $\delta$ feasibility gap). Taking $\delta \to 0$, $T^\star$ becomes impossible to recover from finite samples, whereas $\kappa$ can be estimated effectively.

Lastly, there is an existing line of work on the design of neural estimators for stochastic OT maps [Korotin et al., 2023a,b]. They show that any optimal kernel is the solution of a certain maximin problem, which they approximately solve via a neural net parameterization and stochastic gradient ascent-descent. However, this maximin problem sometimes admits spurious solutions associated with suboptimal maps. In general, these are more empirical works which do not address statistical rates.

## 1.2 Preliminaries

**Notation.** Let $\|\cdot\|$ denote the Euclidean norm on $\mathbb{R}^d$, and $\mathbb{B}^d$ be the $d$-dimensional unit ball. For measurable $S \subseteq \mathbb{R}^d$, write $\mathcal{P}(S)$ for the space of probability measures over $S$ and $\mathrm{diam}(S)$ for its diameter. Let $\mathcal{P}_q(S)$ denote those with finite $q$th moments, and write $\mathcal{N}(x, \Sigma)$ for the multi-variate Gaussian distribution with mean $x \in \mathbb{R}^d$ and covariance $\Sigma \in \mathbb{R}^{d \times d}$. We say that $\mu \in \mathcal{P}(\mathbb{R}^d)$ is $\sigma^2$-sub-Gaussian if $\mathbb{E}_\mu[\exp(\|X\|^2/\sigma^2)] \leq 2$. Write $\mathcal{M}(S)$ for the space of finite signed measures on $S$, equipped with the TV norm $\|\nu\|_{\mathrm{TV}} := \frac{1}{2}|\nu|(S)$, and $\mathcal{M}^+(S)$ for those which are non-negative. Let $\hat{\mu}_n = \frac{1}{n} \sum_{i=1}^n \delta_{X_i}$ be the empirical measure of $n$ i.i.d. samples $X_1, \dots, X_n$ from $\mu$. We write $a \vee b := \max\{a, b\}$, $a \wedge b := \min\{a, b\}$, and $\lesssim_x, \gtrsim_x, \asymp_x$ for (in)equalities up to a constant depending only on $x$ (omitting $x$ for absolute constants).

**Kernels and their composition.** Writing $\mathcal{B}(\mathcal{Y})$ for the Borel subsets of $\mathcal{Y}$, we recall that a Markov kernel $\kappa \in \mathcal{K}(\mathcal{X}, \mathcal{Y})$ is a map $(A, x) : \mathcal{B}(\mathcal{Y}) \times \mathcal{X} \mapsto \kappa_x(A) \in [0, 1]$ which is measurable in $x$ for fixed $A$ and is a probability measure on $\mathcal{Y}$ for fixed $x$. Consequently, given any $\mu \in \mathcal{P}(\mathcal{X})$, the pushforward measure $\kappa_\sharp \mu(\cdot) := \int \kappa_x(\cdot) \mathrm{d}\mu(x)$ is well-defined probability measure on $\mathcal{Y}$. Moreover, fixing any intermediate space $\mathcal{Z} \subseteq \mathbb{R}^d$, kernels $\kappa \in \mathcal{K}(\mathcal{Z}, \mathcal{Y})$ and $\lambda \in \mathcal{K}(\mathcal{X}, \mathcal{Z})$ can be composed to obtain the composite kernel $\kappa \circ \lambda \in \mathcal{K}(\mathcal{X}, \mathcal{Y})$ defined by $(\kappa \circ \lambda)(A|x) := \int \kappa_z(A) \mathrm{d}\lambda_x(z)$.

**Statistical distances and empirical convergence.** We often use the following standard results.

**Fact 1** ($\mathsf{W}_p$-TV comparison). *For $\mu, \nu \in \mathcal{P}(\mathcal{X})$, we have $\mathsf{W}_p(\mu, \nu) \leq \mathrm{diam}(\mathcal{X}) \|\mu - \nu\|_{\mathrm{TV}}^{1/p}$.*

**Lemma 1** ($\mathsf{W}_p$ empirical convergence, Lei, 2020). *If $q > p$ and $\mu \in \mathcal{P}_q(\mathbb{R}^d)$, then $\mathbb{E}[\mathsf{W}_p(\mu, \hat{\mu}_n)] \lesssim_{p,q}$ $\mathbb{E}_\mu[\|X\|^q]^{\frac{1}{q}} n^{-\left[\frac{1}{(2p) \vee d} \wedge \left(\frac{1}{p} - \frac{1}{q}\right)\right]} \log^2(n)$. If $d > q > 2p$, then $\mathbb{E}[\mathsf{W}_p(\mu, \hat{\mu}_n)] \lesssim_{p,q} \mathbb{E}_\mu[\|X\|^q]^{\frac{1}{q}} n^{-\frac{1}{d}}$.*

## 2 Basic Properties of the Error Functional

Before turning to estimation, we establish some fundamental properties of our error functional

$$\mathcal{E}_p(\kappa; \mu, \nu) := \left[ \left( \iint \|x - y\|^p \mathrm{d}\kappa_x(y) \mathrm{d}\mu(x) \right)^{\frac{1}{p}} - \mathsf{W}_p(\mu, \nu) \right]_+ + \mathsf{W}_p(\kappa_\sharp \mu, \nu). \tag{4}$$

This metric is natural because it vanishes for any optimal kernel, namely, $\mathcal{E}_p(\kappa; \mu, \nu) = 0$ if and only if $\kappa$ minimizes (2). Further, it generalizes the existing $L^p$ benchmark, which applies only when an optimal deterministic map $T^\star$ exists.

**Proposition 1** (Relation to $L^p$ loss)**.** *For any map $T : \mathcal{X} \to \mathcal{Y}$ and $T^\star : \mathcal{X} \to \mathcal{Y}$ minimizing (1),* $\mathcal{E}_p(T; \mu, \nu) \leq 2\|T - T^\star\|_{L^p(\mu)}.$

See Appendix A.1 for the proof. In Figure 2, we show how $L^p$ can be arbitrarily large compared to $\mathcal{E}_p$, failing to recognize the performance of map estimates that deviate pointwise from $T^\star$.

**Remark 1** (Optimality vs. feasibility)**.** An alternative version of $\mathcal{E}_p$ weights the feasibility gap by a regularization strength $\lambda$. When $\lambda = 0$, the identity map is optimal, and, as $\lambda \to \infty$, the constant kernel $\kappa(\cdot|x) := \nu$ becomes optimal. Our setting is a natural balance between these two extremes.

**Remark 2** (Reverse $L^2$ comparison)**.** If $p = 2$ and $T^\star$ is the gradient of an $L$-smooth convex potential, we show in Appendix A.2 that $\|T - T_\star\|_{L^2(\mu)}^2 \lesssim (L \vee 1)\, \mathcal{E}_2(T; \mu, \nu)\big(\mathsf{W}_2(\mu, \nu) + \mathcal{E}_2(T; \mu, \nu)\big).$

We also compare $\mathcal{E}_p$ to the Monge gap objective discussed in Section 1. The proof in Appendix A.3 essentially follows by the $\mathsf{W}_p$ triangle inequality.

**Lemma 2** (Comparison to Monge gap)**.** *For the alternative objective $\mathcal{E}'_p$ defined by*

$$\mathcal{E}'_p(\kappa; \mu, \nu)^p := \underbrace{\iint \|x - y\|^p \mathrm{d}\kappa_x(y)\mathrm{d}\mu(x) - \mathsf{W}_p(\mu, \kappa_\sharp \mu)^p}_{\textit{Monge gap}} + \mathsf{W}_p(T_\sharp \mu, \nu)^p,$$

*we have $\mathcal{E}_p \leq 4\mathcal{E}'_p$. For $p = 1$, we have $\mathcal{E}'_1/2 \leq \mathcal{E}_1 \leq 2\mathcal{E}'_1$.*

We now examine stability of $\mathcal{E}_p$ with respect to perturbations of the source and target distributions. These stability results form the backbone for the estimation risk analysis of the estimators proposed in the subsequent sections. Their proofs in Appendix A generally follow via the $L^p$ triangle inequality.

**Lemma 3** (Stability in $\nu$)**.** *Fix $\mu \in \mathcal{P}(\mathcal{X})$ and $\nu, \nu' \in \mathcal{P}(\mathcal{Y})$. For each $\kappa \in \mathcal{K}(\mathcal{X}, \mathcal{Y})$, we have*
$$\big|\mathcal{E}_p(\kappa; \mu, \nu) - \mathcal{E}_p(\kappa; \mu, \nu')\big| \leq 2\,\mathsf{W}_p(\nu, \nu') \leq 2\operatorname{diam}(\mathcal{Y})\|\nu - \nu'\|_{\mathrm{TV}}^{1/p}.$$

The $\mathcal{E}_p$ metric tolerates $\mathsf{W}_p$ perturbations of $\mu$ when $\kappa$ is appropriately Hölder continuous.

**Lemma 4** ($\mathsf{W}_p$ stability in $\mu$)**.** *Fix $\mu, \mu' \in \mathcal{P}(\mathcal{X})$, $\nu \in \mathcal{P}(\mathcal{Y})$, and $\kappa \in \mathcal{K}(\mathcal{X}, \mathcal{Y})$ with $\mathsf{W}_p(\kappa_x, \kappa_{x'}) \leq L\|x - x'\|^\alpha$ for all $x, x' \in \mathcal{X}$, where $0 < \alpha \leq 1$. Setting $\rho = \mathsf{W}_p(\mu', \mu)$, we have*
$$\big|\mathcal{E}_p(\kappa; \mu', \nu) - \mathcal{E}_p(\kappa; \mu, \nu)\big| \leq 2\rho + 2L\rho^\alpha.$$
*In particular, the above holds with $\alpha = 1$ whenever $\kappa$ is induced by a deterministic, $L$-Lipschitz map.*

We can treat TV perturbations of the source measure when $\mathcal{Y}$ is bounded.

**Lemma 5** (TV stability in $\mu$)**.** *Fix $\mu, \mu' \in \mathcal{P}(\mathcal{X})$, $\nu \in \mathcal{P}(\mathcal{Y})$, and kernel $\kappa \in \mathcal{K}(\mathcal{X}, \mathcal{Y})$. Setting $\varepsilon = \|\mu - \mu'\|_{\mathrm{TV}}$, we have*
$$\mathcal{E}_p(\kappa; \mu', \nu) \leq 4\operatorname{diam}(\mathcal{Y})\varepsilon^{1/p} + 10\mathcal{E}_p(\kappa; \mu, \nu) + 6\mathcal{E}_p(\kappa; \mu, \nu)^{\frac{1}{p}}\mathsf{W}_p(\mu, \nu)^{\frac{p-1}{p}}$$
*In particular, we have $\mathcal{E}_1(\kappa; \mu', \nu) \lesssim \operatorname{diam}(\mathcal{Y})\varepsilon + \mathcal{E}_1(\kappa; \mu, \nu)$.*

That is, if $\mu'$ and $\mu$ share substantial mass, and $\kappa$ performs well on $\mu$, then its performance on $\mu'$ under $\mathcal{E}_p$ cannot be substantially worse. In particular, the proof in Appendix A.6 decomposes $\mu = \alpha + \beta$, where $\alpha$ is its shared mass with $\mu'$, and uses that, if $\kappa$ is optimal for the $\mathsf{W}_p(\mu, \nu)$ problem, then it must also be optimal for the $\mathsf{W}_p(\alpha, \kappa_\sharp \alpha)$ sub-problem.

Finally, we consider the behavior of $\mathcal{E}_p$ when evaluated on composite kernels, which we employ in some of the subsequent kernel estimators.

**Lemma 6** (Kernel composition)**.** *Fixing an intermediate space $\mathcal{Z} \subseteq \mathbb{R}^d$, let $\mu \in \mathcal{P}(\mathcal{X})$, $\nu \in \mathcal{P}(\mathcal{Y})$, $\lambda \in \mathcal{K}(\mathcal{X}, \mathcal{Z})$, and $\kappa \in \mathcal{K}(\mathcal{Z}, \mathcal{Y})$. We then bound*
$$\big|\mathcal{E}_p(\kappa \circ \lambda; \mu, \nu) - \mathcal{E}_p(\kappa; \lambda_\sharp \mu, \nu)\big| \leq 2\bigg(\iint \|z - x\|^p \mathrm{d}\lambda_x(z)\mathrm{d}\mu(x)\bigg)^{\frac{1}{p}}.$$
*In particular, for $\lambda$ induced by a deterministic map $f : \mathcal{X} \to \mathcal{Z}$, we have*
$$\big|\mathcal{E}_p(\kappa \circ f; \mu, \nu) - \mathcal{E}_p(\kappa; f_\sharp \mu, \nu)\big| \leq 2\|f - \operatorname{Id}\|_{L^p(\mu)}.$$

This can be applied iteratively to analyze the composition of many kernels, peeling off one at a time.

# 3 Finite-Sample Estimation and Computation

Under $\mathcal{E}_p$, we can perform kernel estimation without regularity or existence of a Brenier map. We analyze one estimator based on EOT using $\mathsf{W}_p$ stability (Lemma 4) and another based on rounding using TV stability (Lemma 5). The former is closer to existing estimators, but the latter achieves sharper rates under milder assumptions. Fixing $n \geq 3$, we have i.i.d. samples $X_1, \ldots, X_n \sim \mu \in \mathcal{P}(\mathcal{X})$ and $Y_1, \ldots, Y_n \sim \nu \in \mathcal{P}(\mathcal{Y})$, whose empirical measures we denote by $\hat{\mu}_n$ and $\hat{\nu}_n$ respectively.

**Entropic kernel estimator.** As a warm-up, we recall the EOT problem, defined for $\tau > 0$ by

$$S_{p,\tau}(\mu, \nu) := \inf_{\pi \in \Pi(\mu,\nu)} \iint \|x - y\|^p \mathrm{d}\pi(x,y) + \tau \mathsf{D}_{\mathrm{KL}}(\pi \| \mu \otimes \nu), \tag{5}$$

where the Kullback-Leibler (KL) divergence is $\mathsf{D}_{\mathrm{KL}}(\mu \| \nu) := \int \log\left(\frac{d\mu}{d\nu}\right) \mathrm{d}\mu$ if $\mu \ll \nu$ and $+\infty$ otherwise. This objective is strictly convex due to the KL penalty, so a unique optimal coupling always exists (supposing that the value is finite). Most solvers for EOT use its equivalent dual formulation:

$$S_{p,\tau}(\mu, \nu) = \sup_{\substack{f \in L^1(\mu) \\ g \in L^1(\nu)}} \int f \mathrm{d}\mu + \int g \, \mathrm{d}\nu - \tau \iint e^{(f(x)+g(x)-\|x-y\|^p)/\tau} \mathrm{d}\mu(x) \mathrm{d}\nu(y) + \tau. \tag{6}$$

The primal and dual structure of EOT is well-studied (see, e.g., Cuturi, 2013, Genevay et al., 2019). In particular, if $\pi_\tau$ minimizes (5), then there exists maximizers $f_\tau, g_\tau$ for (6), termed *entropic potentials*, such that the conditional *entropic kernel* $\pi_\tau(\cdot|x)$ can be written as

$$\begin{aligned}
\mathrm{d}\pi_\tau(y|x) &= \exp\big((f_\tau(x) + g_\tau(y) - \|x-y\|^p)/\tau\big) \, \mathrm{d}\nu(y) \\
&= \frac{\exp\big((g_\tau(y) - \|x-y\|^p)/\tau\big) \, \mathrm{d}\nu(y)}{\int \exp\big((g_\tau(y') - \|x-y'\|^p)/\tau\big) \, \mathrm{d}\nu(y')}.
\end{aligned} \tag{7}$$

By this result, we may assume that the entropic kernel is defined over all $x \in \mathbb{R}^d$. We show that, if $\mathrm{diam}(\mathcal{X} \cup \mathcal{Y})$ is bounded, the empirical entropic kernel achieves a vanishing $\mathcal{E}_p$ error.

**Theorem 1** (Entropic kernel estimator). *Assume $\mathcal{X}, \mathcal{Y} \subseteq [0,1]^d$ and set $\tau = d^{p/4} n^{-1/(2d \vee 4)} \log n$. Let $\hat{\pi}_{\tau,n}$ be the optimal coupling for $S_{p,\tau}(\hat{\mu}_n, \hat{\nu}_n)$. Then, the conditional kernel $\hat{\kappa}_n$ defined by $(\hat{\kappa}_n)_x = \pi_{\tau,n}(\cdot|x)$ satisfies $\mathbb{E}[\mathcal{E}_p(\hat{\kappa}_n; \mu, \nu)] \lesssim_{p,d} n^{-1/(2pd \vee 4p)} \log^2(n)$.*

The proof in Appendix B.1 has three steps. First, we control $\mathcal{E}_p(\hat{\kappa}_n; \hat{\mu}_n, \hat{\nu}_n) = \widetilde{O}_d(\tau^{1/p})$ using a known bound of Genevay et al. [2019]. Then, using the support constraint and the softmax form in (7), we bound the TV Lipschitz constant of $\hat{\kappa}_n$ by $O_d(\tau^{-1})$, which implies the kernel is $\mathsf{W}_p$ Hölder continuous with exponent $1/p$ and constant $O_d(\tau^{-1/p})$. Finally, we apply Lemmas 3 and 4 to bound

$$\mathcal{E}_p(\hat{\kappa}_n; \mu, \nu) \leq \mathcal{E}_p(\hat{\kappa}_n; \hat{\mu}_n, \hat{\nu}_n) + O_d((\rho/\tau)^{1/p}) + O(\rho) = \widetilde{O}_d(\tau^{1/p} + (\rho/\tau)^{1/p} + \rho),$$

where $\rho = \mathsf{W}_p(\hat{\mu}_n, \mu) \vee \mathsf{W}_p(\hat{\nu}_n, \nu)$. Applying Lemma 1 to bound $\rho$ and tuning $\tau$ gives the theorem. Note that $\tau$ controls a bias-variance trade-off ($\tau \to 0$ overfits, while $\tau \to \infty$ blurs out all structure).

To understand the quality of this bound, we compare to Brenier map estimation with $p = 2$. Here, the conditional kernel is usually converted into a deterministic map $\hat{T}_n$ via barycentric projection, which sends $x$ to the mean of $Y \sim \hat{\pi}_{\tau,n}(\cdot|x)$. Existing work has derived a variety of $L^2$ estimation guarantees for this estimator with respect to the Brenier map $T^\star$. In particular, Pooladian and Niles-Weed [2021] show that $\mathbb{E}\|\hat{T}_n - T^\star\|_{L^2(\mu)} \lesssim_d n^{-(\alpha+1)/[4(d+\alpha+1)]} \log n$ if $T^\star \in \mathcal{C}^{\alpha+1}$ for $1 < \alpha \leq 3$ and $\nabla T^\star$ has eigenvalues bounded from above and below. For the sake of comparison, we can take a formal limit as $\alpha \to 0$ (although not covered by their theory) to obtain a rate of $n^{-1/(4d+4)}$, which is always worse than our $n^{-1/(4d \vee 8)}$ rate (which does not require $p = 2$ nor the existence of $T^\star$).

**Improved rounding estimator.** While guarantees for the entropic kernel estimator from Theorem 1 are compelling, we can achieve sharper rates using the following rounding estimator, via an analysis based on TV stability of $\mathcal{E}_p$. The estimator is specified by an accuracy $\delta \geq 0$, a trimming radius $R > 0$, a partition $\mathcal{P}$ of $\mathbb{R}^d$, and a collection of centers $C_\mathcal{P} = \{c_P\}_{P \in \mathcal{P}}$ such that each $c_P \in P$. This induces a rounding function $r_\mathcal{P} : \mathbb{R}^d \to C_\mathcal{P}$ which, for each $P \in \mathcal{P}$, maps $x \in P$ to $c_P$. Given empirical measures $\hat{\mu}_n$ and $\hat{\nu}_n$, we proceed as follows:

1. Round $\hat{\mu}_n$ onto $\mathcal{P}$, taking $\mu'_n = (r_\mathcal{P})_\sharp \hat{\mu}_n$

2. Compute a preliminary kernel $\bar{\kappa}_n \in \mathcal{K}(C_\mathcal{P}, \mathrm{supp}(\hat{\nu}_n))$ which pushes $\mu'_n$ onto $\hat{\nu}_n$ and is near-optimal for the $W_p$ problem, satisfying $\iint \|x - y\|^p \mathrm{d}\bar{\kappa}_n(y|x)\mathrm{d}\mu'_n(x) \leq W_p(\mu'_n, \hat{\nu}_n)^p + \delta$.

3. Return kernel $\hat{\kappa}_n = \bar{\kappa}_n \circ r_\mathcal{P}$, which, given $x \in \mathbb{R}^d$, rounds it to $r_\mathcal{P}(x)$ before applying $\bar{\kappa}_n$.

For a simple choice of $\mathcal{P}$, this procedure achieves low $\mathcal{E}_p$ error when $\mu$ and $\nu$ are sub-Gaussian. With a more complex partition, we can support $\mu$ with only bounded $2p$th moments.

**Theorem 2** (Rounding estimator). *Let $\mu, \nu$ be 1-sub-Gaussian, and take $\mathcal{P}$ as the regular partition of $\mathbb{R}^d$ into cubes of side length $r$. Then, for $R$, $r$, and $\delta$ tuned independently of $\mu$ and $\nu$, we have $\mathbb{E}[\mathcal{E}_p(\hat{\kappa}_n; \mu, \nu)] = \widetilde{O}_{p,d}(n^{-1/(d+2p)})$. For an alternative, non-uniform partition, this guarantee still holds if the sub-Gaussianity assumption on $\mu$ is relaxed to $\mathbb{E}_\mu[\|X\|^{2p}] \leq 1$. In both cases, computation is dominated by Step 2 which, if implemented via an EOT solver, runs in time $O((C_\infty + d)n^{2+o_d(1)})$, where $C_\infty = \max_{i,j} \|X_i - Y_j\|$.*

This improved $n^{-1/(d+2p)}$ rate is near-optimal; indeed, when $\mu$ is a point mass, the problem reduces to estimation of $\nu$ under $W_p$, for which there are existing minimax lower bounds of order $n^{-1/(d \vee 2p)}$ [Singh and Póczos, 2018] (see Appendix B.2 for full details). We also note that the tail bounds on $\mu$ and $\nu$ can be weakened further under our analysis, but not without worsening the rate.

We sketch the proof when $\mu$ is sub-Gaussian, $\delta = 0$, and $\mathrm{diam}(\mathcal{Y}) \leq 1$; see Appendix B.3 for the full derivation. We first show, for the cubic partition $\mathcal{P}$ with side length $r$, that $\mu'_n = (r_\mathcal{P})_\sharp \hat{\mu}_n$ and $\mu' = (r_\mathcal{P})_\sharp \mu$ converge in TV at rate $\sqrt{r^{-d}/n}$. Here, $r^{-d}$ arises as a bound on the number of relevant partition blocks. We then bound

$$\begin{aligned}
\mathcal{E}_p(\hat{\kappa}_n; \mu, \nu) &= \mathcal{E}_p(\bar{\kappa}_n \circ r_\mathcal{P}; \mu, \nu) \\
&\leq \mathcal{E}_p(\bar{\kappa}_n; \mu', \nu) + \sqrt{d}r && \text{(Lemma 6)} \\
&\lesssim W_p(\nu, \hat{\nu}_n) + (nr^d)^{-\frac{1}{2p}} + \sqrt{d}r && \text{(Lemmas 3 and 5)}
\end{aligned}$$

Applying Lemma 1 and tuning $r$ gives the theorem. The general sub-Gaussian case follows by a similar argument. If $\mu$ only has bounded $2p$th moments, we can still achieve TV convergence at a comparable rate by employing a partition whose bins increase in size away from the origin.

**Remark 3** (One-dimensional refinements). Given the gap between our $n^{-1/(d+2p)}$ upper bound and the $n^{-1/(d \vee 2p)}$ lower bound of Singh and Póczos [2018], it is natural to ask if the lower bound can be improved. At least when $d = 1$, this is impossible. In Appendix B.4, we improve the rate from Theorem 2 to $n^{-1/2}$ when $d = 1$, using stability of $\mathcal{E}_p$ under the Kolmogorov-Smirnov metric.

# 4 Improved Statistical Guarantees with Hölder Continuous Optimal Kernels

For $p = 2$, many existing works assume the existence of a Brenier map $T^\star$ whose gradient has eigenvalues bounded from above and below (in particular, such a $T^\star$ is Lipschitz). For example, Balakrishnan and Manole [2025] show that a nearest-neighbor estimator achieves $\mathbb{E}\|\hat{T}_n - T^\star\|_{L^2(\mu)} = \widetilde{O}(n^{-1/(d \vee 4)})$, matching the $W_2$ empirical convergence rate. Of course, by Proposition 1, this guarantee also holds under $\mathcal{E}_2$. In this section, we treat general $p \geq 1$ under the related but distinctly weaker assumption that $W_p(\mu, \nu)$ admits an optimal kernel which is Hölder continuous under $W_p$.

**Assumption 1.** There exists $L \geq 1$, $\alpha \in (0, 1]$, and an optimal kernel $\kappa^\star \in \mathcal{K}(\mathcal{X}, \mathcal{Y})$ for $W_p(\mu, \nu)$ such that $W_p(\kappa_x^\star, \kappa_{x'}^\star) \leq L\|x - x'\|^\alpha$ for all $x, x' \in \mathcal{X}$.

Here, we propose an estimator based on Wasserstein distributionally robust optimization (WDRO):

$$\hat{\kappa}_{\mathrm{DRO}}^\rho[\hat{\mu}, \hat{\nu}] := \underset{\kappa \in \mathcal{K}(\mathbb{R}^d, \mathcal{Y})}{\arg\min} \; \underset{\mu' \in \mathcal{P}(\mathbb{R}^d): W_p(\mu', \hat{\mu}) \leq \rho}{\sup} \; \mathcal{E}_p(\kappa; \mu', \hat{\nu}),$$

where $\hat{\mu}, \hat{\nu}$ are any proxies for $\mu, \nu$ (potentially their empirical measures). This estimator is computationally inefficient but allows us to better understand the statistical limits of estimation.

**Theorem 3** (WDRO estimator). *Under Assumption 1, suppose $W_p(\hat{\mu}, \mu) \leq \rho_\mu$ and $W_p(\hat{\nu}, \nu) \leq \rho_\nu$. Then the estimate $\hat{\kappa} = \hat{\kappa}_{\mathrm{DRO}}^{2\rho_\mu}[\hat{\mu}, \hat{\nu}]$ achieves $\mathcal{E}_p(\hat{\kappa}; \mu, \nu) \lesssim L\rho_\mu^\alpha + \rho_\mu + \rho_\nu$.*

*Proof.* Using the WDRO problem structure and our stability lemmas, we bound

$$\mathcal{E}_p(\hat{\kappa}; \mu, \nu) \leq \mathcal{E}_p(\hat{\kappa}; \mu, \hat{\nu}) + 2\rho_\nu \qquad\qquad\qquad\text{(Lemma 3)}$$

$$\leq \sup_{\mu': \mathsf{W}_p(\mu', \hat{\mu}) \leq \rho_\mu} \mathcal{E}_p(\hat{\kappa}; \mu', \hat{\nu}) + 2\rho_\nu \qquad\qquad (\mathsf{W}_p(\mu, \hat{\mu}) \leq \rho_\mu)$$

$$\leq \sup_{\mu': \mathsf{W}_p(\mu', \hat{\mu}) \leq \rho_\mu} \mathcal{E}_p(\kappa_\star; \mu', \hat{\nu}) + 2\rho_\nu \qquad\qquad (\text{optimality of } \hat{\kappa})$$

$$\leq \sup_{\mu' \in \mathcal{P}(\mathbb{R}^d): \mathsf{W}_p(\mu', \mu) \leq 2\rho_\mu} \mathcal{E}_p(\kappa_\star; \mu', \hat{\nu}) + 2\rho_\nu \qquad (\mathsf{W}_p \text{ triangle inequality})$$

$$\leq 4L\rho_\mu^\alpha + 4\rho_\mu + 2\rho_\nu, \qquad\qquad\qquad\qquad\text{(Lemma 4)}$$

as desired. $\qquad\qquad\qquad\qquad\qquad\qquad\qquad\qquad\qquad\qquad\qquad\qquad\qquad\qquad\square$

This gives an information-theoretic reduction from kernel estimation under $\mathcal{E}_p$ to estimation of $\mu$ and $\nu$ under $\mathsf{W}_p$, i.e., if we can estimate $\mu$ up to error $\rho_\mu$ and $\nu$ up to error $\rho_\nu$, then we can find a kernel with error $O(L\rho_\mu^\alpha + \rho_\mu + \rho_\nu)$. Focusing on the Lipschitz case with $\alpha = 1$, we first apply Theorem 3 using the plug-in estimators $\mu = \hat{\mu}_n, \nu = \hat{\nu}_n$.

**Corollary 1** (Plug-in estimators)**.** *Under Assumption 1 with $\alpha = 1$ and $L = O(1)$, suppose $\mu, \nu \in \mathcal{P}_{2p}(\mathbb{R}^d)$. Then taking $\hat{\mu} = \hat{\mu}_n$ and $\hat{\nu} = \hat{\nu}_n$, $\rho$ can be tuned to achieve $\mathcal{E}_p(\hat{\kappa}_{\mathrm{DRO}}^\rho; \mu, \nu) = \widetilde{O}_{p,d}(n^{-1/(d \vee 2p)})$ with probability 0.9, which is minimax optimal up to logarithmic factors.*

Plugging in $p = 2$, we recover the $L^2$ rate of Balakrishnan and Manole [2025]; however, this result holds under our significantly weaker assumption and for general $p \geq 1$. If further $\mu$ and $\nu$ are compactly supported with smooth densities, we can employ wavelet-based distribution estimators (see, e.g., Weed and Berthet [2019], Manole et al. [2024]) to attain faster rates.

**Corollary 2** (Wavelet estimators)**.** *Under Assumption 1 with $\alpha = 1$ and $L = O(1)$, suppose that $\mu, \nu \in \mathcal{P}([0,1]^d)$ admit Lebesgue densities $f, g \in \mathcal{C}^s([0,1]^d)$. Then taking $\hat{\mu}$ and $\hat{\nu}$ as appropriate wavelet-based estimators, one can tune $\delta$ to achieve $\mathcal{E}_p(\hat{\kappa}; \mu, \nu) = \widetilde{O}_{p,d}(n^{-[(1+s/p)/(d+s) \wedge 1/(2p)]})$ with probability 0.9, which is minimax optimal up to logarithmic factors.*

Balakrishnan and Manole [2025] also reduce map estimation to distribution estimation, so they prove a variety of similar guarantees. However, unlike our derivation, their analysis relies crucially on the structure of the Brenier map as the gradient of a sufficiently regular convex potential.

**Remark 4** (Lipschitz regularization)**.** Wasserstein DRO is known to be closely related to Lipschitz regularization (see, e.g., Gao et al., 2022). So perhaps expectedly, one can show for $p = \alpha = 1$ that the guarantees of $\hat{\kappa}_{\mathrm{DRO}}$ are matched by the estimator which minimizes the regularized empirical risk $\kappa \mapsto \mathcal{E}_1(\kappa; \hat{\mu}, \hat{\nu}) + \lambda \operatorname{Lip}(x \mapsto \kappa_x; \mathsf{W}_1)$. For deterministic map estimation, González-Sanz et al. [2022] considered related neural estimators that enforced Lipschitz constraints on the estimated map. In general, minimizing the unregularized empirical risk $\mathcal{E}_1$, or, by Lemma 2, the corresponding Monge gap objective, achieves good rates whenever the obtained minimizer has a small Lipschitz constant (whether this arises due to explicit constraints or implicit optimization bias). This gives a partial explanation for the empirical success of the Monge gap regularizer for neural map estimation.

## 5 Robust Estimation with Adversarial Corruptions

The previous sections allowed us to handle sampling error under $\mathcal{E}_p$. We now address local and global adversarial perturbations of the data points with minimal technical overhead, thanks to the strong stability properties of $\mathcal{E}_p$ in both TV and $\mathsf{W}_p$.

**Formal corruption model and assumptions.** As discussed in the introduction, TV and $\mathsf{W}_p$ perturbations have historically been studied separately in robust statistics to model outliers and adversarial examples, respectively, with the former dating back to Huber [1964]. Our work adopts a recent combined model permitting both local and global perturbations of the input data [Nietert et al., 2023b]. Here, clean i.i.d. data from the unknown distributions $\mu \in \mathcal{P}(\mathcal{X})$ and $\nu \in \mathcal{P}(\mathcal{Y})$ are first nudged by small local perturbations (namely, in Wasserstein distance with budget $\rho \geq 0$) and then partially overwritten by global outliers (in TV, with allowed fraction $\varepsilon \in [0,1]$). More precisely, letting $X_1, \ldots, X_n \overset{\text{i.i.d.}}{\sim} \mu$ and $Y_1, \ldots, Y_n \overset{\text{i.i.d.}}{\sim} \nu$ denote the clean samples, we observe $\tilde{X}_1, \ldots, \tilde{X}_n \in \mathcal{X}$ and

$\tilde{Y}_1, \ldots, \tilde{Y}_n \in \mathcal{Y}$ such that $\frac{1}{n} \sum_{i \in S} \|\tilde{X}_i - X_i\|^p \vee \frac{1}{n} \sum_{i \in T} \|\tilde{X}_i - X_i\|^p \leq \rho^p$ for some $S, T \subseteq [n]$ with $|S|, |T| \geq (1 - \varepsilon)n$.

Write $\hat{\mu}_n, \tilde{\mu}_n$ and $\hat{\nu}_n, \tilde{\nu}_n$ for the clean and corrupted empirical measures for $\mu$ and $\nu$, respectively. We further suppose that $\mu$ is 1-sub-Gaussian and $\mathcal{Y} \subseteq [0, 1]^d$. These assumptions can be relaxed at the cost of estimation complexity, as in Section 3. We impose them to focus on the new aspects of adversarial robustness without distractions.

An initial idea is to combine the Section 3 approaches, since the entropic kernel used $\mathsf{W}_p$ stability and the rounding estimator used TV stability. This is viable, however our entropic kernel analysis requires that $\mathcal{X} \cup \mathcal{Y}$ is bounded. To avoid this, we employ a similar approach to the rounding estimator, but replace deterministic rounding with Gaussian convolution. Defining the kernel $N_x^\sigma = \mathcal{N}(x, \sigma^2 I_d)$ and letting $\kappa_p^\star[\alpha \to \beta]$ denote (any) optimal kernel for the $\mathsf{W}_p(\alpha, \beta)$ problem, we consider

$$\hat{\kappa}_{\text{conv}}^\sigma[\tilde{\mu}_n, \tilde{\nu}_n] := \kappa_p^\star[N_\sharp^\sigma \tilde{\mu}_n \to \tilde{\nu}_n] \circ N^\sigma.$$

That is, we find an optimal kernel for the convolved $\mathsf{W}_p(N_\sharp^\sigma \tilde{\mu}_n, \tilde{\nu}_n)$ problem and compose it with the convolution kernel. The initial convolution of $\tilde{\mu}_n$ ensures that the inner kernel is defined over all of $\mathbb{R}^d$, potentially outside the support of $\tilde{\mu}_n$. The subsequent composition ensures that the outer kernel is sufficiently continuous, as needed to apply Lemma 4. We prove the following in Appendix D.

**Theorem 4** (Robust estimation guarantee). *Under the setting above, we have*

$$\mathbb{E}[\mathcal{E}_p(\hat{\kappa}_{\text{conv}}^\sigma; \mu, \nu)] \lesssim \sqrt{d}\varepsilon^{\frac{1}{p}} + \sqrt{d}\rho^{\frac{1}{p+1}} + \rho + O_{p,d}(n^{-\frac{1}{d+2p}}),$$

*for tuned $\sigma = \sigma(\rho, d, p)$. Also, the naïve estimator $(\hat{\kappa}_{\text{null}})_x \equiv \delta_0$ satisfies $\mathcal{E}_p(\hat{\kappa}_{\text{null}}; \mu, \nu) \leq \sqrt{d}$. By selecting between the two estimators according to which bound is smaller, we achieve an error bound of $\left(\sqrt{d}\varepsilon^{\frac{1}{p}} + \sqrt{d}\rho^{\frac{1}{p+1}} + \rho + O_{p,d}(n^{-\frac{1}{d+2p}})\right) \wedge \sqrt{d}$. Moreover, up to constants, no estimator can achieve worst-case expected error less than $\left(\sqrt{d}\varepsilon^{\frac{1}{p}} + d^{1/4}\rho^{1/2} + n^{-\frac{1}{d\vee 2p}}\right) \wedge \sqrt{d}$.*

The upper bound follows by a remarkably straightforward application of our stability lemmas. For the $d^{1/4}\rho^{1/2}$ term in the LB, we construct a pair of instances (with all distributions supported on two points) which are indistinguishable from $\rho$-corrupted samples and such that no kernel achieves error $o(d^{1/4}\rho^{1/2})$ on both. Interestingly, this $\sqrt{\rho}$ dependence rules out a lossless reduction from estimation under $\mathcal{E}_p$ to distribution estimation under $\mathsf{W}_p$. That is, our rounding estimator from Section 3 achieves $\mathcal{E}_p = \widetilde{O}(n^{-1/(d+2p)})$ but the guarantee that $\mathsf{W}_p(\hat{\mu}_n, \mu) \vee \mathsf{W}_p(\hat{\nu}_n, \nu) = \widetilde{O}(n^{-1/(d\vee 2p)})$ alone cannot imply a rate faster than $\widetilde{O}(n^{-1/(2d\vee 4p)})$. Finally, although the convolved OT problem for our estimator may not be efficiently solvable, we show in Appendix D that an additional rounding step enables efficient computation, mirroring the proof of Theorem 2.

## 6 Experiments

To empirically validate our theory, we run experiments in two synthetic settings with OT maps whose irregularities limit the utility of the $L^p$ objective and prevent application of existing theory. For Setting A, we fix $\mu$ and $\nu$ as uniform discrete measures over $N = 2000$ points, obtained as i.i.d. samples from $\text{Unif}(\{0\} \times [0, 1]^{d-1})$ and $\frac{1}{2}\text{Unif}(\{-1\} \times [0, 1]^{d-1}) + \frac{1}{2}\text{Unif}(\{1\} \times [0, 1]^{d-1})$, respectively. In the $N \to \infty$ limit, the optimal kernel satisfies $\kappa_{(0, x_{2:d})}^\star = \text{Unif}(\{(-1, x_{2:d}), (1, x_{2:d})\})$. For our discrete $\mu$ and $\nu$, there is an optimal deterministic map $T^\star$ induced by a permutation, but it is highly oscillatory. For Setting B, we set $\mu$ and $\nu$ as discrete distributions over $N$ samples from $\text{Unif}([-1, 1]^d)$ and $f_\sharp \text{Unif}([-1, 1]^d)$, respectively, where $f(x) = x + (\text{sign}(x_1), \ldots, \text{sign}(x_d))$ pushes each orthant of the cube away from the origin. Here, the OT map is discontinuous but Lipschitz within each orthant.

Now, for each setting and sample size $n \in \{10, 20, \ldots, 100\}$, we take $n$ i.i.d. samples from $\mu$ and $\nu$ and compute the $p = 1$ nearest-neighbor map estimate $\hat{T}_n^{\text{NN}}$ [Manole et al., 2024] and the rounding kernel estimate $\hat{\kappa}_n^{\text{round}}$ (Section 3). (Specifically, the NN estimator first computes an optimal $\mathsf{W}_1$ map $\bar{T}_n$ from $\hat{\mu}_n$ to $\hat{\nu}_n$. Then, $\hat{T}_n^{\text{NN}}$ maps each $x \in \mathbb{R}^d$ to the image of its nearest source point under $\bar{T}_n$.) We then compute the $L^1$ error $\|\hat{T}_n^{\text{NN}} - T^\star\|_{L^1(\mu)}$ and the $\mathcal{E}_1$ errors $\mathcal{E}_1(\hat{T}_n^{\text{NN}}; \mu, \nu), \mathcal{E}_1(\hat{\kappa}_n^{\text{round}}; \mu, \nu)$. Since $\mu$ and $\nu$ are discrete, these can be computed using finite sums and the default Python Optimal Transport solver [Flamary et al., 2021]. Repeating this process for $K = 100$ iterations, we compute mean errors for each sample size and dimension $d \in \{3, 5, 10\}$, along with bootstrapped 10% and 90% quantiles (via 1000 bootstrap resamples). In Figure 3 (left), we compare the $\mathcal{E}_1$ vs $L^1$

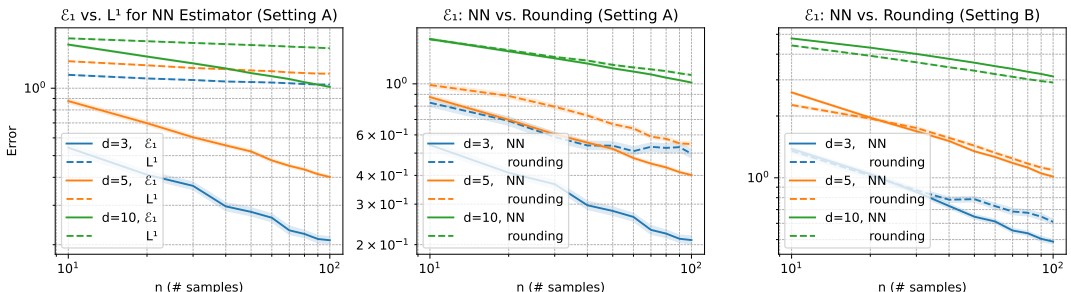

Figure 3: $\mathcal{E}_1$ and $L^1$ performance of nearest-neighbor and rounding estimators in two settings.

performance of the NN estimator under Setting A (where the latter is well-defined since each $\hat{T}_n^{\mathrm{NN}}$ is a deterministic map). As expected, $L^1$ performance is quite poor, with error always greater than 1. Although we currently lack formal guarantees for the NN estimator (and our setting lies outside of existing theory), it achieves strong $\mathcal{E}_1$ performance, with faster rates in lower dimensions. In Figure 3 (center), we compare the NN and rounding estimators under $\mathcal{E}_1$, the latter enjoying formal guarantees by Theorem 2. Empirically, the NN estimator performs better, but this gap diminishes in high dimensions. We suspect that low-dimensional performance of the rounding estimator is more sensitive to its side-length hyperparameter, which we have simply set to $n^{-1/(d+2)}$ as per the proof of Theorem 2. Finally, in Figure 3 (right) we turn to Setting B, again comparing $\hat{T}_n^{\mathrm{NN}}$ and $\hat{\kappa}_n^{\mathrm{round}}$ under $\mathcal{E}_1$ and observing similar trends. We note that all experiments were performed on an M1 MacBook Air with 16GB RAM and 8 CPU cores. See Appendix E for full experiment details and two additional experiments (one with larger parameter settings, but no bootstrapping, and one in two dimensions, so that our estimator can be visualized).

**Remark 5** (Neural map estimation). Given the connections between $\mathcal{E}_p$ and the Monge gap objective discussed in Sections 1 and 2, one could consider training a neural map estimator to minimize an empirical $\mathcal{E}_p$ objective, perhaps after approximating the $\mathsf{W}_p$ terms via EOT. However, while both $\mathcal{E}_p$ and the Monge gap objective nullify on optimal maps, they behave quite differently when far from optimality. Indeed, gradients of the feasibility gap term in $\mathcal{E}_p$ push towards the identity map (since it achieves the minimum transport cost of zero), while gradients of the Monge gap push towards the much larger set of $c$-cyclically monotone maps. In preliminary tests, we found that the Monge gap objective led to significantly more stable training dynamics, which we attribute to this difference. Thus, we maintain our recommendation of $\mathcal{E}_p$ as an evaluation metric, enabling provable error guarantees under weaker assumptions, rather than a training objective for neural map estimation. Still, we hope that our analysis under $\mathcal{E}_p$ might inspire new regularization methods in the future.

## 7  Discussion

This work proposed a novel error metric $\mathcal{E}_p$ which broadens the scope of OT map estimation research to support stochastic maps, sidestepping existence, uniqueness, and regularity issues faced by existing approaches and treating $p \neq 2$. We developed an efficient rounding estimator with near-optimal rates under $\mathcal{E}_p$ and characterized the minimax rate for Lipschitz continuous kernels. Our analysis extends naturally to adversarial corruptions, and our theory is supported by numerical simulations.

There are two clear open questions. First, what is the minimax finite-sample risk for estimation under $\mathcal{E}_p$, say for $\mu, \nu \in \mathcal{P}([0,1]^d)$? We have established that the correct rate lies between $n^{-1/(d \vee 2p)}$ and $n^{-1/(d+2p)}$. The slower rate mirrors that attained by bounding $\mathbb{E}[\mathsf{W}_p(\hat{\mu}_n, \mu)]$ without analyzing sampling error at multiple geometric scales. Can a multi-scale approach extend to kernel estimation and improve the current upper bound in Section 3? Second, under the setting of Section 4 with $\alpha = 1$, where there exists an optimal Lipschitz kernel, can a computationally efficient estimator achieve the optimal $n^{-1/(d \vee 2p)}$ rate? Our experiments demonstrate strong empirical performance of the NN estimator in varied settings, so it seems to be a promising candidate to attain such a guarantee.

Finally, our objective can naturally be extended to many OT variants, including EOT, weak OT [Gozlan et al., 2017], conditional OT [Hosseini et al., 2025], and adapted OT [Bartl et al., 2024]. Adapting our toolkit of stability lemmas to such settings is an interesting direction for future work.

## Acknowledgments and Disclosure of Funding

Z. Goldfeld is partially supported by NSF grants CCF-2046018, DMS-2210368, and CCF-2308446, and the IBM Academic Award.

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

# A Proofs for Section 2

## A.1 Proof of Proposition 1

Clearly, $\mathcal{E}_p(\kappa; \mu, \nu) = 0$ if $\kappa$ minimizes (2). On the other hand, if $\mathcal{E}_p(\kappa, \mu, \nu) = 0$, then $\kappa_\sharp \mu = \nu$. Thus, $\kappa$ is feasible for (2) with optimal objective value, i.e., it is a minimizer.

Further, if $T^\star$ is an optimal map, then $W_p(\mu, \nu) = \|T^\star - \mathrm{Id}\|_{L^p(\mu)}$ and $T^\star_\sharp \mu = \nu$. We thus bound

$$
\begin{aligned}
\mathcal{E}_p(T; \mu, \nu) &= \left[\|T - \mathrm{Id}\|_{L^p(\mu)} - W_p(\mu, \nu)\right]_+ + W_p(T_\sharp \mu, \nu) \\
&= \left[\|T - \mathrm{Id}\|_{L^p(\mu)} - \|T^\star - \mathrm{Id}\|_{L^p(\mu)}\right]_+ + W_p(T_\sharp \mu, T^\star_\sharp \mu) \\
&\leq 2\|T - T^\star\|_{L^p(\mu)},
\end{aligned}
$$

as desired. $\qquad\square$

## A.2 Reverse $L^2$ comparison (Remark 2)

Suppose that there exists a unique Brenier map of the form $T^\star = \nabla\varphi$, where $\varphi : \mathbb{R}^d \to \mathbb{R}$ is convex and twice differentiable such that $H\varphi \preceq LI_d$. Fixing any map $T : \mathcal{X} \to \mathcal{Y}$, we abbreviate $\varepsilon = \mathcal{E}_2(T; \mu, \nu)$. By the definition of $\mathcal{E}_2$, we have $W_2(T_\sharp \mu, \nu) \leq \varepsilon$. Let $\lambda \in \mathcal{K}(\mathcal{Y}, \mathcal{Y})$ be a kernel which achieves this bound, and take $\kappa = \lambda \circ T$. By construction, we have $\kappa_\sharp \mu = \nu$ and

$$
\left(\iint \|y - x\|^2 \mathrm{d}\kappa(y|x)\mathrm{d}\mu(x)\right)^{\frac{1}{2}} - W_2(\mu, \nu) \leq \left(\iint \|T(x) - x\|^2\mathrm{d}\mu(x)\right)^{\frac{1}{2}} - W_2(\mu, \nu) + \varepsilon
$$
$$
\leq 2\varepsilon.
$$

Consequently, we have

$$
\iint \|y - x\|^2 \mathrm{d}\kappa(y|x)\mathrm{d}\mu(x) - W_2(\mu, \nu)^2 \leq 2\varepsilon\left(\left(\iint \|y - x\|^2\mathrm{d}\kappa(y|x)\mathrm{d}\mu(x)\right)^{\frac{1}{2}} + W_2(\mu, \nu)\right)
$$
$$
\leq 2\varepsilon \cdot (2W_2(\mu, \nu) + 2\varepsilon)
$$

Thus, by Proposition 3.1 of Li and Nochetto [2021], we have

$$
\iint \|y - T^\star(x)\|^2\mathrm{d}\kappa(y|x)\mathrm{d}\mu(x) \leq L\left(\iint \|y - x\|^2\mathrm{d}\kappa(y|x)\mathrm{d}\mu(x) - W_2(\mu, \nu)^2\right)
$$
$$
\leq 4L\varepsilon \cdot (W_2(\mu, \nu) + \varepsilon).
$$

Finally, we bound

$$
\begin{aligned}
\|T - T_\star\|_{L^2(\mu)} &\leq \left(\iint \|y - T^\star(x)\|^2\mathrm{d}\kappa(y|x)\mathrm{d}\mu(x)\right)^{\frac{1}{2}} + \varepsilon \\
&\leq \sqrt{4L\varepsilon \cdot (W_2(\mu, \nu) + \varepsilon)} + \varepsilon \\
&\lesssim \sqrt{(L \vee 1) \cdot \varepsilon \cdot (W_2(\mu, \nu) + \varepsilon)},
\end{aligned}
$$

as desired.

## A.3 Proof of Lemma 2

First, we bound $\mathcal{E}_p$ in terms of $\mathcal{E}'_p$, computing

$$\mathcal{E}_p(\kappa; \mu, \nu) = \left[\left(\iint \|y - x\|^p \mathrm{d}\kappa_x(y) \mathrm{d}\mu(x)\right)^{\frac{1}{p}} - W_p(\mu, \nu)\right]_+ + W_p(T_\sharp \mu, \nu)$$

$$\leq \left[\left(\iint \|y - x\|^p \mathrm{d}\kappa_x(y) \mathrm{d}\mu(x)\right)^{\frac{1}{p}} - W_p(\mu, T_\sharp \mu)\right]_+ + 2W_p(T_\sharp \mu, \nu)$$

$$\leq \left[\iint \|y - x\|^p \mathrm{d}\kappa_x(y) \mathrm{d}\mu(x) - W_p(\mu, T_\sharp \mu)^p\right]_+^{\frac{1}{p}} + 2W_p(T_\sharp \mu, \nu)$$

$$= \left[\iint \|y - x\|^p \mathrm{d}\kappa_x(y) \mathrm{d}\mu(x) - W_p(\mu, T_\sharp \mu)^p\right]_+^{\frac{1}{p}} + 2W_p(T_\sharp \mu, \nu)$$

$$\leq 2^{2-\frac{1}{p}} \left(\iint \|y - x\|^p \mathrm{d}\kappa_x(y) \mathrm{d}\mu(x) - W_p(\mu, T_\sharp \mu)^p + W_p(T_\sharp \mu, \nu)^p\right)^{\frac{1}{p}}$$

$$\leq 2^{2-\frac{1}{p}} \mathcal{E}'_p(\kappa; \mu, \nu),$$

where the second inequality uses that $[a^{1/p} - b^{1/p}]_+ \leq [a - b]_+^{1/p}$ for all $a, b \geq 0$, and the penultimate inequality uses that $\ell_1 \leq 2^{1-1/p} \ell_p$ in $\mathbb{R}^2$. This implies the claimed bound of $\mathcal{E}_p \leq 4\mathcal{E}'_p$. When $p = 1$, the above gives $\mathcal{E}_1 \leq 2\mathcal{E}'_1$, and we similarly bound

$$\mathcal{E}'_1(\kappa; \mu, \nu) = \iint \|y - x\| \mathrm{d}\kappa_x(y) \mathrm{d}\mu(x) - W_1(\mu, T_\sharp \mu) + W_1(T_\sharp \mu, \nu)$$

$$= \left[\iint \|y - x\| \mathrm{d}\kappa_x(y) \mathrm{d}\mu(x) - W_1(\mu, T_\sharp \mu)\right]_+ + W_1(T_\sharp \mu, \nu)$$

$$\leq \left[\iint \|y - x\| \mathrm{d}\kappa_x(y) \mathrm{d}\mu(x) - W_1(\mu, \nu)\right]_+ + 2W_1(T_\sharp \mu, \nu)$$

$$\leq 2\mathcal{E}_1(\kappa; \mu, \nu)$$

as desired. $\qquad \square$

## A.4 Proof of Lemma 3

We simply bound

$$\left|\mathcal{E}_p(\kappa; \mu, \nu) - \mathcal{E}_p(\kappa; \mu, \nu')\right| \leq |W_p(\mu, \nu) - W_p(\mu, \nu')| + |W_p(\kappa_\sharp \mu, \nu) - W_p(\kappa_\sharp \mu, \nu')|$$

$$\leq 2W_p(\nu, \nu')$$

$$\leq 2\operatorname{diam}(\mathcal{Y}) \|\nu - \nu'\|_{\mathrm{TV}},$$

where the final inequality uses Fact 1. $\qquad \square$

## A.5 Proof of Lemma 4

While the key ideas of this proof are straightforward, measurability issues require some care (we encourage the reader to skip such details on an initial read). In what follows, we equip all spaces of distributions with the weak topology and always employ Borel measurability. By the definition of a Markov kernel, $x \in \mathcal{X} \mapsto \kappa_x(A)$ is a measurable function for each measurable $A \subseteq \mathcal{Y}$. Thus, $(x, x') \in \mathcal{X}^2 \mapsto (\kappa_x(A), \kappa_{x'}(B))$ is measurable for fixed, measurable $A, B \subseteq \mathcal{Y}$, implying that $(x, x') \in \mathcal{X}^2 \mapsto (\kappa_x, \kappa_{x'}) \in \mathcal{P}(\mathcal{Y})^2$ is measurable. Therefore, by Theorem 3.0.8 of Toneian [2019], there exists a measurable map $(x, x') \in \mathcal{X}^2 \mapsto \gamma_{x,x'} \in \Pi(\kappa_x, \kappa_{x'})$ such that $\gamma_{x,x'}$ is an OT plan for $W_p(\kappa_x, \kappa_{x'})$ for all $x, x' \in \mathcal{X}$.

Now, let $\pi_0 \in \Pi(\mu, \mu')$ be an OT plan for $W_p(\mu, \mu')$, and define the joint law $\pi$ by $\pi(A \times B \times C \times D) := \iint_{A \times B} \gamma_{x,x'}(C \times D) \mathrm{d}\pi_0(x, x')$, which is well-defined due to the measurability argument above. Taking $(X, X', Y, Y') \sim \pi$, our construction ensures the following:

- $X \sim \mu$ and $X' \sim \mu'$ such that $\mathbb{E}[\|X - X'\|^p] = \rho^p$,

- $Y \sim \kappa_X$ and $Y' \sim \kappa_{X'}$ such that $\mathbb{E}[\|Y - Y'\|^p | X, X'] = \mathsf{W}_p(\kappa_X, \kappa_{X'})^p$.

Consequently, we bound

$$
\begin{aligned}
\mathbb{E}[\|Y - Y'\|^p] &= \mathbb{E}\big[\mathbb{E}\big[\|Y - Y'\|^p \big| X, X'\big]\big] \\
&= \mathbb{E}\big[\mathsf{W}_p(\kappa_X, \kappa_{X'})^p\big] \\
&\leq \mathbb{E}[L^p \|X - X'\|^{\alpha p}] && \text{(Hölder continuity of } \kappa) \\
&\leq L^p \, \mathbb{E}[L\|X - X'\|^p]^{\alpha} && \text{(Jensen's inequality, } 0 < \alpha \leq 1) \\
&= L^p \rho^{\alpha p}.
\end{aligned}
$$

Moreover, using Minkoswki's inequality, we compute

$$
\Big|\mathbb{E}[\|Y - X\|^p]^{\frac{1}{p}} - \mathbb{E}[\|Y' - X'\|^p]^{\frac{1}{p}}\Big| \leq \mathbb{E}[\|X - X'\|^p]^{\frac{1}{p}} + \mathbb{E}[\|Y - Y'\|^p]^{\frac{1}{p}} \leq \rho + L\rho^{\alpha}.
$$

Finally, we bound $|\mathsf{W}_p(\mu, \nu) - \mathsf{W}_p(\mu', \nu)| \leq \mathsf{W}_p(\mu, \mu') \leq \rho$ and

$$
|\mathsf{W}_p(\kappa_\sharp \mu, \nu) - \mathsf{W}_p(\kappa_\sharp \mu', \nu)| \leq \mathsf{W}_p(\kappa_\sharp \mu, \kappa_\sharp \mu') \leq \mathbb{E}[\|Y - Y'\|^p]^{\frac{1}{p}} \leq L\rho^{\alpha}.
$$

The definition of $\mathcal{E}_p$ and these bounds give the lemma. $\qquad\square$

## A.6  Proof of Lemma 5

To show this result, we prove a slightly more general lemma.

**Lemma 7.** *Fix $\mu, \mu' \in \mathcal{P}(\mathcal{X})$, $\nu \in \mathcal{P}(\mathcal{Y})$, and kernel $\kappa \in \mathcal{K}(\mathcal{X}, \mathcal{Y})$ with $\iint \|y - x\|^p \mathrm{d}\kappa_x(y)\mathrm{d}\mu(x) \leq \mathsf{W}_p(\mu, \kappa_\sharp \mu)^p + \tau^p$ and $\mathsf{W}_p(\kappa_\sharp \mu, \nu) \leq \tau$ for some $\tau \geq 0$. Then, setting $\varepsilon = \|\mu - \mu'\|_{\mathrm{TV}}$, we have $\mathcal{E}_p(\kappa; \mu', \nu) \leq 3 \operatorname{diam}(\mathcal{Y})\varepsilon^{1/p} + 3\tau$.*

*Proof.* In what follows, we encourage the reader to focus on the $p = 1$ case, where computations are more direct. Write $\varepsilon = \|\mu - \mu'\|_{\mathrm{TV}}$. By the TV bound, there exist $\alpha, \beta, \gamma \in \mathcal{M}_+(\mathcal{X})$ with $\gamma(\mathcal{X}) = 1 - \varepsilon$ and $\alpha(\mathcal{X}) = \beta(\mathcal{X}) = \varepsilon$ such that $\mu = \gamma + \alpha$ and $\mu' = \gamma + \beta$.

First, we note that $\kappa$ must perform well on $\gamma$. Specifically, we have

$$
\begin{aligned}
\iint \|x - y\|^p &\mathrm{d}\kappa_x(y)\mathrm{d}\gamma(x) \\
&= \iint \|x - y\|^p \mathrm{d}\kappa_x(y)\mathrm{d}\mu(x) - \iint \|x - y\|^p \mathrm{d}\kappa_x(y)\mathrm{d}\alpha(x) && (\gamma = \mu - \alpha) \\
&\leq \mathsf{W}_p(\mu, \kappa_\sharp \mu)^p + \tau^p - \mathsf{W}_p(\alpha, \kappa_\sharp \alpha)^p && \text{(error bound for } \kappa, \text{ def. of } \mathsf{W}_p) \\
&\leq \mathsf{W}_p(\gamma, \kappa_\sharp \gamma)^p + \mathsf{W}_p(\alpha, \kappa_\sharp \alpha)^p + \tau^p - \mathsf{W}_p(\alpha, \kappa_\sharp \alpha)^p && (\mu = \gamma + \alpha) \\
&= \mathsf{W}_p(\gamma, \kappa_\sharp \gamma)^p + \tau^p \\
&\leq (\mathsf{W}_p(\gamma, \kappa_\sharp \gamma) + \tau)^p. && (\ell_p \leq \ell_1)
\end{aligned}
$$

Now, letting $\kappa'$ be an optimal kernel for the $\mathsf{W}_p(\mu', \nu)$ problem and writing $D = \operatorname{diam}(\mathcal{Y})$, we have

$$\left(\iint \|y - x\|^p \mathrm{d}\kappa_x(y)\mathrm{d}\mu'(x)\right)^{\frac{1}{p}} - \mathsf{W}_p(\mu', \nu)$$

$$= \left(\iint \|y - x\|^p \mathrm{d}\kappa_x(y)\mathrm{d}\mu'(x)\right)^{\frac{1}{p}} - \left(\iint \|y - x\|^p \mathrm{d}\kappa'_x(y)\mathrm{d}\mu'(x)\right)^{\frac{1}{p}} \qquad \text{(optimality of } \kappa')$$

$$= \left(\iint \|y - x\|^p \mathrm{d}\kappa_x(y)\mathrm{d}\gamma(x) + \iint \|y - x\|^p \mathrm{d}\kappa_x(y)\mathrm{d}\beta(x)\right)^{\frac{1}{p}} \qquad (\mu' = \gamma + \beta)$$

$$\quad - \left(\iint \|y - x\|^p \mathrm{d}\kappa'_x(y)\mathrm{d}\gamma(x) + \iint \|y - x\|^p \mathrm{d}\kappa'_x(y)\mathrm{d}\beta(x)\right)^{\frac{1}{p}}$$

$$\leq \left(\left[\left(\iint \|y - x\|^p \mathrm{d}\kappa_x(y)\mathrm{d}\gamma(x)\right)^{\frac{1}{p}} - \left(\iint \|y - x\|^p \mathrm{d}\kappa'_x(y)\mathrm{d}\gamma(x)\right)^{\frac{1}{p}}\right]_+^p \right.$$

$$\quad + \left.\left[\left(\iint \|y - x\|^p \mathrm{d}\kappa_x(y)\mathrm{d}\beta(x)\right)^{\frac{1}{p}} - \left(\iint \|y - x\|^p \mathrm{d}\kappa'_x(y)\mathrm{d}\beta(x)\right)^{\frac{1}{p}}\right]_+^p \right)^{\frac{1}{p}}$$

$$\leq \left(\left[\mathsf{W}_p(\gamma, \kappa_\sharp\gamma) + \tau - \mathsf{W}_p(\gamma, \kappa'_\sharp\gamma)\right]_+^p \right.$$

$$\quad + \left.\left[\left(\iint \|y - x\|^p \mathrm{d}\kappa_x(y)\mathrm{d}\beta(x)\right)^{\frac{1}{p}} - \left(\iint \|y - x\|^p \mathrm{d}\kappa'_x(y)\mathrm{d}\beta(x)\right)^{\frac{1}{p}}\right]_+^p \right)^{\frac{1}{p}}$$

$$\leq \left(\left(\mathsf{W}_p(\kappa_\sharp\gamma, \kappa'_\sharp\gamma) + \tau\right)^p + \iiint \|y - y'\|^p \mathrm{d}\kappa_x(y)\mathrm{d}\kappa'_x(y')\mathrm{d}\beta(x)\right)^{\frac{1}{p}}$$

$$\leq \left(\left(\mathsf{W}_p(\kappa_\sharp\gamma, \kappa'_\sharp\gamma) + \tau\right)^p + \varepsilon D^p\right)^{\frac{1}{p}} \qquad \text{(Fact 1)}$$

$$\leq \mathsf{W}_p(\kappa_\sharp\gamma, \kappa'_\sharp\gamma) + \tau + D\varepsilon^{\frac{1}{p}}. \qquad (\ell_p \leq \ell_1)$$

The first inequality uses that $(A^p + B^p)^{1/p} - (a^p + b^p)^{1/p} \leq ([A - a]_+^p + [B - b]_+^p)^{1/p}$, which can be obtained by rearranging the $\ell_p$ triangle inequality and using that $A = [A - a]_+ + A \wedge a$. The second inequality uses the previous bound and the fact that $\kappa'$ is feasible for the $\mathsf{W}_p(\gamma, \kappa'_\sharp\gamma)$ problem. The third uses the $\mathsf{W}_p$ triangle inequality and Minkoswki's inequality.

We next bound $\mathsf{W}_p(\kappa_\sharp\gamma, \kappa'_\sharp\gamma)$. Let $\pi \in \Pi(\kappa_\sharp\mu, \nu)$ be an optimal plan for $\mathsf{W}_p(\kappa_\sharp\mu, \nu)$ and define $\lambda \in (1 - \varepsilon)\mathcal{P}(\mathcal{Y})$ by $\lambda(\cdot) = \int \pi(\cdot|x)\mathrm{d}\gamma(x)$. By construction, $\mathsf{W}_p(\kappa_\sharp\gamma, \lambda) \leq \mathsf{W}_p(\kappa_\sharp\mu, \nu) \leq \tau$. Moreover, both $\lambda$ and $\kappa'_\sharp\gamma$ are submeasures of $\nu$ with mass $1 - \varepsilon$, and so they must share common mass at least $1 - 2\varepsilon$. This implies that their TV distance is at most $\varepsilon$, and so Fact 1 gives that

$$\mathsf{W}_p(\kappa_\sharp\gamma, \kappa'_\sharp\gamma) \leq \tau + \mathsf{W}_p(\lambda, \kappa'_\sharp\gamma) \leq \tau + D\varepsilon^{\frac{1}{p}}. \tag{8}$$

Thus, the previous bound on the optimality gap can be tightened to

$$\left(\iint \|y - x\|^p \mathrm{d}\kappa_x(y)\mathrm{d}\mu'(x)\right)^{\frac{1}{p}} - \mathsf{W}_p(\mu', \nu) \leq \tau + 2D\varepsilon^{\frac{1}{p}}.$$

Similarly, we bound the feasibility gap by

$$\mathsf{W}_p(\kappa_\sharp\mu', \nu)^p = \mathsf{W}_p(\kappa_\sharp\gamma + \kappa_\sharp\beta, \kappa'_\sharp\gamma + \kappa'_\sharp\beta)^p \qquad (\mu' = \gamma + \beta, \kappa'_\sharp\gamma = \nu)$$

$$\leq \mathsf{W}_p(\kappa_\sharp\gamma, \kappa'_\sharp\gamma)^p + \mathsf{W}_p(\kappa_\sharp\beta, \kappa'_\sharp\beta)^p \qquad \text{(joint convexity of } \mathsf{W}_p^p)$$

$$\leq \tau^p + (\tau + D\varepsilon^{\frac{1}{p}})^p \qquad \text{(Fact 1 and Eq. 8)}$$

$$\leq (2\tau + D\varepsilon^{\frac{1}{p}})^p. \qquad (\ell_p \leq \ell_1)$$

Combining, we have that $\mathcal{E}_p(\kappa; \mu', \nu) \leq 3\tau + 3D\varepsilon^{1/p}$, as desired. $\qquad\square$

## A.7 Proof of Lemma 6

First, we note that $\kappa_\sharp(\lambda_\sharp\mu) = (\kappa \circ \lambda)_\sharp\mu$ by the definition of kernel composition. This implies that the two feasibility gaps coincide. Moreover, by Minkowski's inequality, we have

$$\left| \left( \iint \|y-z\|^p \mathrm{d}\kappa_z(y)\mathrm{d}(\lambda_\sharp\mu)(z) \right)^{\frac{1}{p}} - \left( \iint \|y-x\|^p \mathrm{d}(\kappa \circ \lambda)_x(y)\mathrm{d}\mu(x) \right)^{\frac{1}{p}} \right|$$
$$\leq \left( \iint \|z-x\|^p \mathrm{d}\lambda_x(z)\mathrm{d}\mu(x) \right)^{\frac{1}{p}}$$

and

$$|\mathsf{W}_p(\lambda_\sharp\mu, \nu) - \mathsf{W}_p(\mu,\nu)| \leq \mathsf{W}_p(\mu, \lambda_\sharp\mu) \leq \left( \iint \|z-x\|^p \mathrm{d}\lambda_x(z)\mathrm{d}\mu(x) \right)^{\frac{1}{p}}.$$

Combining these two error bounds gives the lemma. □

# B Proofs for Section 3

## B.1 Proof of Theorem 1

By the support constraint, our cost $\|x-y\|^p$ is $pd^{(p-1)/2}$-Lipschitz over $\mathcal{X} \times \mathcal{Y}$. Thus, by Theorem 1 of Genevay et al. [2019], we have

$$S_{p,\tau}(\hat{\mu}_n, \hat{\nu}_n) \leq \mathsf{W}_p(\hat{\mu}_n, \hat{\nu}_n)^p + 2\tau d \log\left(e^2 pd^{p/2-1}\tau^{-1}\right).$$

Since $\hat{\pi}_{\tau,n}$ achieves the left hand side above and KL divergence is non-negative, we have

$$\iint \|x-y\|^p \mathrm{d}\hat{\kappa}_n(y|x)\mathrm{d}\hat{\mu}_n(x) \leq \mathsf{W}_p(\hat{\mu}_n, \hat{\nu}_n)^p + 2\tau d \log\left(e^2 pd^{p/2-1}\tau^{-1}\right).$$

Taking $p$th roots and noting that $(\hat{\kappa}_n)_\sharp\hat{\mu}_n = \hat{\nu}_n$, this implies that

$$\mathcal{E}_p(\hat{\kappa}_n; \hat{\mu}_n, \hat{\nu}_n) \leq \left[ 2\tau d \log\left(e^2 pd^{p/2-1}\tau^{-1}\right) \right]^{\frac{1}{p}} \leq 4(\tau d)^{\frac{1}{p}} \log(e^2 d\tau^{-1}).$$

Now, by (7), $(\hat{\kappa}_n)_x$ is obtained by applying softmax to $v(x) := \left( (g_\tau(Y_i) - \|x-Y_i\|^p)/\tau \right)_{i=1}^n \in \mathbb{R}^n$. Since the $\ell_1, \ell_\infty$ Lipschitz constant of the softmax operation is $\leq 1$, we have

$$\|(\hat{\kappa}_n)_x - (\hat{\kappa}_n)_{x'}\|_{\mathrm{TV}} \leq \frac{1}{2}\|v(x) - v(x')\|_\infty \leq \frac{1}{2}pd^{(p-1)/2}\tau^{-1}\|x-x'\|_2$$

for all $x, x' \in [0,1]^d$. Thus, by Fact 1, $\hat{\kappa}_n$ is Hölder continuous under $\mathsf{W}_p$ with exponent $1/p$ and constant $2\sqrt{d}\tau^{-1/p}$. Applying Lemma 4 now gives

$$\begin{aligned}
\mathcal{E}_p(\hat{\kappa}_n; \mu, \nu) &\leq \mathcal{E}_p(\hat{\kappa}_n; \mu, \hat{\nu}_n) + \mathsf{W}_p(\hat{\nu}_n, \nu) \\
&\leq \mathcal{E}_p(\hat{\kappa}_n; \hat{\mu}_n, \hat{\nu}_n) + 2\mathsf{W}_p(\mu, \hat{\mu}_n) + 4\sqrt{d}\,\mathsf{W}_p(\mu, \hat{\mu})^{\frac{1}{p}}\tau^{-\frac{1}{p}} + \mathsf{W}_p(\nu, \hat{\nu}_n) \\
&\leq 4(\tau d)^{\frac{1}{p}} \log(e^2 d\tau^{-1}) + 4\sqrt{d}\,\mathsf{W}_p(\mu, \hat{\mu})^{\frac{1}{p}}\tau^{-\frac{1}{p}} + 2\mathsf{W}_p(\mu, \hat{\mu}_n) + \mathsf{W}_p(\nu, \hat{\nu}_n).
\end{aligned}$$

Taking expectations, applying Lemma 1, and plugging in $\tau$ gives the theorem. □

## B.2 Minimax Lower Bound under Sampling

Fix $\mu = \delta_0$, so that the constant kernel $\kappa^\star$ defined by $\kappa_x^\star \equiv \nu$ is optimal. Note that the error $\mathcal{E}_p(\kappa; \mu, \nu)$ of any kernel $\kappa$ is thus lower bounded by the feasibility gap $\mathsf{W}_p(\kappa_0, \nu)$. Since we only observe $n$ i.i.d. samples from $\nu \in \mathcal{P}([0,1]^d)$, any upper bound on an estimator for this problem instance also gives an upper bound for $n$-sample distribution estimation of $\nu$ under $\mathsf{W}_p$. However, the minimax lower bound of Singh and Póczos [2018] implies that no distribution estimator can achieve $\mathsf{W}_p$ error less than $n^{-1/(2p\vee d)}$ for all $\nu \in \mathcal{P}([0,1]^d)$.

### B.3 Proof of Theorem 2

We start with some helpful lemmas.

**Lemma 8** (Rigollet [2015], Theorem 1.14). *Let $\mu \in \mathcal{P}(\mathbb{R})$ be 1-sub-Gaussian. Then, for $X_1, \ldots, X_n$ sampled i.i.d. from $\mu$, we have $\max_{i=1,\ldots,n} X_i \leq \sqrt{2\log(n/\delta)}$ with probability at least $1 - \delta$.*

**Lemma 9.** *Let $\mu \in \mathcal{P}(\mathbb{R}^d)$ be 1-sub-Gaussian and let $\mathcal{P}$ denote the regular partition of $\mathbb{R}^d$ into cubes of side-length $r > 0$. Then, for any choice of rounding map $r_{\mathcal{P}}$, we have*

$$\mathbb{E}[\|(r_{\mathcal{P}})_{\sharp}(\hat{\mu}_n - \mu)\|_{\mathrm{TV}}] = \widetilde{O}\left(\sqrt{\frac{5^d r^{-d}}{n}}\right).$$

*Proof.* Let $B$ denote a ball of radius $R = \sqrt{2\log(n)}$ centered at the origin, so that $\mu(B) \geq 1 - 1/n$ by Lemma 8. Write $\mathcal{P}_R$ for the subset of partition blocks $P \in \mathcal{P}$ which intersect $B$, and note that $|\mathcal{P}_R| \leq \mathrm{vol}(B) r^{-d} \leq (3R/r)^d$. We then bound

$$
\begin{aligned}
\mathbb{E}[\|(r_{\mathcal{P}})_{\sharp}(\hat{\mu}_n - \mu)\|_{\mathrm{TV}}] &= \frac{1}{2}\mathbb{E}\left[\sum_{P \in \mathcal{P}} |(\hat{\mu}_n - \mu)(P)|\right] \\
&= \frac{1}{2}\mathbb{E}\left[\sum_{P \in \mathcal{P}_R} |(\hat{\mu}_n - \mu)(P)| + \sum_{P \in \mathcal{P} \backslash \mathcal{P}_R} |(\hat{\mu}_n - \mu)(P)|\right] \\
&\lesssim \sqrt{\frac{|\mathcal{P}_R|}{n}} + \mathbb{E}\left[\sum_{P \in \mathcal{P} \backslash \mathcal{P}_R} \hat{\mu}_n(P) + \mu(P)\right] \\
&\lesssim \sqrt{\frac{|\mathcal{P}_R|}{n}} + \mu(\mathbb{R}^d \backslash B) \\
&\lesssim \sqrt{\frac{(3\sqrt{2\log(n)})^d r^{-d}}{n}} + \frac{1}{n} \\
&= \widetilde{O}\left(\sqrt{\frac{5^d r^{-d}}{n}}\right),
\end{aligned}
$$

as desired. $\qquad\square$

**Lemma 10.** *There exists a partition $\mathcal{P}$ parameterized by $\delta > 0$ such that, for all $\mu \in \mathcal{P}(\mathbb{R}^d)$ with $\mathbb{E}_\mu[\|X\|^{p+1}] \leq 1$ and any rounding map $r_{\mathcal{P}}$, we have $\mathbb{E}[\|(r_{\mathcal{P}})_{\sharp}(\hat{\mu}_n - \mu)\|_{\mathrm{TV}}] = \widetilde{O}(\sqrt{\delta^{-d}/n})$ and $\|r_{\mathcal{P}} - \mathrm{Id}\|_{L^p(\mu)} \lesssim \delta$.*

*Proof.* Let $X_0$ be a minimal $(3\delta)$-covering of the unit ball, denoted $S_0$. In particular, this implies that $|X_0| \leq \delta^{-d}$. Now, take $\mathcal{P}_0$ to be the Voronoi partition of $S_0$ induced by $X_0$, so that $\mathcal{P}_0$ has at most $\delta^{-d}$ cells of diameter at most $6\delta$. Then for each integer $i > 0$, set $S_i := 2^i S_0 \backslash 2^{i-1} S_0$, and let $\mathcal{P}_i$ be the dilated partition $\{(2^i P) \cap S_i : P \in \mathcal{P}_0\}$. By construction, $|\mathcal{P}_i| \leq \delta^{-d}$ and each $P \in \mathcal{P}_i$ has diameter at most $2^i \cdot 6\delta$, for all $i \geq 0$. Moreover, by Markov's inequality, we have

$$\mu(S_i) \leq \Pr_\mu\big(\|X\| > 2^{i-1}\big) = \Pr_\mu\big(\|X\|^{p+1} > 2^{(p+1)(i-1)}\big) \leq 2^{(1-i)(p+1)}$$

for each $i > 0$. We thus bound

$$\mathbb{E}[\|(r_{\mathcal{P}})_\sharp(\mu - \hat{\mu}_n)\|_{\mathrm{TV}}] = \mathbb{E}\left[\sum_{i=0}^{\infty}\sum_{P \in \mathcal{P}_i} |(\mu - \hat{\mu}_n)(P)|\right]$$

$$\leq \sum_{i=0}^{\infty}\sum_{P \in \mathcal{P}_i} \sqrt{\mathrm{Var}_{\mu^{\otimes n}}[\hat{\mu}_n(P)]}$$

$$\leq \frac{1}{\sqrt{n}} \cdot \sum_{i=0}^{\infty}\sum_{P \in \mathcal{P}_i} \sqrt{\mu(P)}$$

$$\leq \frac{1}{\sqrt{n}} \cdot \sum_{i=0}^{\infty} \sqrt{|\mathcal{P}_i|\mu(S_i)}$$

$$\leq (\delta^d n)^{-\frac{1}{2}} \cdot \left(1 + \sum_{i=1}^{\infty} 2^{\frac{1-i}{2}}\right)$$

$$\lesssim (\delta^d n)^{-\frac{1}{2}}.$$

Similarly, we bound

$$\|r_{\mathcal{P}} - \mathrm{Id}\|_{L^p(\mu)} \leq \left(\sum_{i=0}^{\infty}\sum_{P \in \mathcal{P}_i} \mu(P)\,\mathrm{diam}(P)^p\right)^{\frac{1}{p}}$$

$$\leq \left(\sum_{i=0}^{\infty} (2^i \cdot 6\delta)^p \mu(S_i)\right)^{\frac{1}{p}}$$

$$\lesssim \left(\sum_{i=0}^{\infty} (2^i \delta)^p 2^{(1-i)(p+1)}\right)^{\frac{1}{p}}$$

$$= \delta\left(\sum_{i=0}^{\infty} 2^{p+1-i}\right)^{\frac{1}{p}}$$

$$\lesssim \delta,$$

as desired. □

We now prove the theorem. First, note that for $D = \sqrt{4\log(n)}$, we have $\max_{i=1,\dots,n}\|Y_i\| \leq DS$ with probability at least $1/n$, by Lemma 8. Now, for a general partition $\mathcal{P}$, we bound

$$\mathcal{E}_p(\hat{\kappa}_n; \mu, \nu) = \mathcal{E}_p(\bar{\kappa}_n \circ r_{\mathcal{P}}; \mu, \nu)$$

$$\leq \mathcal{E}_p(\bar{\kappa}_n; \mu', \nu) + 2\|r_{\mathcal{P}} - \mathrm{Id}\|_{L^p(\mu)} \qquad \text{(Lemma 6)}$$

$$\leq \mathcal{E}_p(\bar{\kappa}_n; \mu', \hat{\nu}_n) + 2\|r_{\mathcal{P}} - \mathrm{Id}\|_{L^p(\mu)} + \mathsf{W}_p(\nu, \hat{\nu}_n) \qquad \text{(Lemma 3)}$$

$$\lesssim \|(r_{\mathcal{P}})_\sharp(\mu - \hat{\mu}_n)\|_{\mathrm{TV}}^{1/p} \cdot \mathrm{diam}(\mathrm{supp}(\hat{\nu}_n)) + \delta^{\frac{1}{p}} + \|r_{\mathcal{P}} - \mathrm{Id}\|_{L^p(\mu)} + \mathsf{W}_p(\nu, \hat{\nu}_n),$$

where the last inequality follows by Lemma 7 and our choice of $\bar{\kappa}_n$. Applying this bound for the regular cube partition and taking expectations, we bound

$$\mathbb{E}[\mathcal{E}_p(\hat{\kappa}_n; \mu, \nu)] \lesssim D\,\mathbb{E}[\|(r_{\mathcal{P}})_\sharp(\mu - \hat{\mu}_n)\|_{\mathrm{TV}}]^{1/p} + \frac{1}{n} + \delta^{\frac{1}{p}} + \sqrt{d}r + \mathsf{W}_p(\nu, \hat{\nu}_n)$$

$$\lesssim \widetilde{O}\left(\frac{5^d r^{-d}}{n}\right)^{\frac{1}{2p}} + \delta^{\frac{1}{p}} + \sqrt{d}r + \widetilde{O}_p\left(n^{-\frac{1}{d\vee 2p}}\right). \qquad \text{(Lemmas 1, 8 and 9)}$$

Taking $r = n^{-1/(d+2p)}$, we obtain $\mathbb{E}[\mathcal{E}_p(\hat{\kappa}_n; \mu, \nu)] = \widetilde{O}_{p,d}(n^{-1/(d+2p)}) + \delta^{1/p}$. The same rate is obtained under bounded $2p$th moments by using the alternative partition from Lemma 10. Thus, to achieve the desired rate, it suffices to solve the preliminary OT problem to accuracy $\delta = n^{-p/(d+2p)}$.

Computational complexity is dominated by this OT computation. The source and target distributions are both supported on $n$ points, and we require accuracy $\delta = n^{-p/(d+2p)}$. Computing the relevant

cost matrix requires time $O(n^2 d)$. Using a state of the art OT solver based on entropic OT (e.g., Luo et al., 2023) gives a running time of $O(C_\infty n^2/\delta) = O(C_\infty n^{2+p/(d+2p)})$, where $C_\infty$ is the largest distance between a source point and a target point.

## B.4 One-Dimensional Refinements (Remark 3)

In one dimension, OT maps can be expressed concisely in terms of CDFs; in particular, if $\mu$ and $\nu$ have strictly increasing CDFs $F_\mu$ and $F_\nu$, respectively, then the map $T^\star(x) = F_\nu^{-1}(F_\mu(x))$ solves the $\mathsf{W}_p(\mu, \nu)$ problem for all $p \geq 1$. As a result, many OT-based inference tasks become more analytically tractable when $d = 1$, including map estimation. In fact, minor adjustments to folklore techniques imply that the optimal risk of $n^{-1/(2p)}$ is achievable when $d = 1$. We now provide a clean derivation of this risk bound using the Kolmogorov-Smirnov (KS) distance.

The KS distance is a useful alternative to the TV metric in one dimension, defined via $\|\mu - \nu\|_{\mathrm{KS}} := \sup_{t \in \mathbb{R}} |(\mu - \nu)((-\infty, t])| = \|F_\mu - F_\nu\|_\infty$. We always have $\|\mu - \nu\|_{\mathrm{KS}} \leq \|\mu - \nu\|_{\mathrm{TV}}$, since $\|\mu - \nu\|_{\mathrm{TV}}$ can alternatively be expressed as $\sup_{A \text{ meas.}} |(\mu - \nu)(A)|$. A comparison with $\mathsf{W}_p$ mirroring Fact 1 is direct.

**Lemma 11** ($\mathsf{W}_p$-KS comparison). *For $\mu, \nu \in \mathcal{P}([0, D])$, we have $\mathsf{W}_p(\mu, \nu) \leq D\|\mu - \nu\|_{\mathrm{KS}}^{1/p}$.*

*Proof.* Writing $F, G$ for the CDFs of $\mu$ and $\nu$, with generalized inverses $F^{-1}$ and $G^{-1}$, we bound

$$
\begin{aligned}
\mathsf{W}_p(\mu, \nu)^p &= \int_0^1 |F^{-1}(u) - G^{-1}(u)|^p \mathrm{d}u \\
&\leq D^{p-1} \int_0^1 |F^{-1}(u) - G^{-1}(u)| \mathrm{d}u \\
&= D^{p-1} \int_0^D |F(x) - G(x)| \mathrm{d}x \\
&\leq D^p \|\mu - \nu\|_{\mathrm{KS}}.
\end{aligned}
$$

Taking $p$th roots gives the statement. $\qquad\square$

The KS distance admits useful empirical convergence guarantees not shared by the TV distance.

**Fact 2** (KS empirical convergence, Massart, 1990). *For all $\mu \in \mathcal{P}(\mathbb{R})$, $\mathbb{E}[\|\mu - \hat{\mu}_n\|_{\mathrm{KS}}] \leq 1/\sqrt{n}$.*

Moreover, for fixed $\mu$ and $\nu$, there exists an optimal kernel for $\mathsf{W}_p(\mu, \nu)$ (namely, based on CDFs as above), which is near-optimal for all $\mu'$ in a KS neighborhood of $\mu$, as shown next.

**Lemma 12** (KS corruptions in $\mu$). *For $\mathcal{X}, \mathcal{Y} \subseteq \mathbb{R}$, fix $\mu \in \mathcal{P}(\mathcal{X})$ and $\nu \in \mathcal{P}(\mathcal{Y})$. There exists an optimal kernel $\kappa^\star \in \mathcal{K}(\mathcal{X}, \mathcal{Y})$ for the $\mathsf{W}_p(\mu, \nu)$ problem such that, for all $\mu' \in \mathcal{P}(\mathcal{X})$, we have*

$$
\mathcal{E}_p(\kappa^\star; \mu', \nu) \lesssim \mathrm{diam}(\mathcal{Y}) \|\mu - \mu'\|_{\mathrm{KS}}^{1/p}.
$$

*Proof.* Write $F, F', G$, for the CDFs of $\mu$, $\mu'$, and $\nu$, respectively, and let $\varepsilon = \|\mu - \mu'\|_{\mathrm{KS}} = \|F - F'\|_\infty$. Write $D = \mathrm{diam}(\mathcal{Y})$ and suppose without loss of generality that $\mathcal{Y} = [0, D]$. For now, suppose further that $1/\varepsilon = 3M$ is a multiple of 3 (without loss of generality) and that $F$ is continuous (which will be relaxed). We consider the kernel induced by the map $T^\star = G^{-1} \circ F$, where $G^{-1}$ is the generalized inverse of $G$ with $G^{-1}(q)$ defined as as 0 for $q \leq 0$ and $D$ for $q \geq 1$. In particular, we compare $T^\star$ with the optimal kernel $G^{-1} \circ F'$ for the $\mathsf{W}_p(\mu', \nu)$ problem, bounding

$$
\begin{aligned}
\int_{\mathcal{X}} \left| G^{-1}(F(x)) - G^{-1}(F'(x)) \right|^p \mathrm{d}F'(x) &\leq \int_{\mathcal{X}} \left| G^{-1}(F'(x) \pm \varepsilon) - G^{-1}(F'(x)) \right|^p \mathrm{d}F'(x) \\
&= \int_0^1 \left| G^{-1}(u \pm \varepsilon) - G^{-1}(u) \right|^p \mathrm{d}u \\
&= \sum_{i=0}^{3M-1} \int_{i\varepsilon}^{(i+1)\varepsilon} \left| G^{-1}(u \pm \varepsilon) - G^{-1}(u) \right|^p \mathrm{d}u \\
&\leq \varepsilon \sum_{i=0}^{3M-1} \left[ G^{-1}((i+2)\varepsilon) - G^{-1}((i-1)\varepsilon) \right]^p.
\end{aligned}
$$

Here, the first equality uses that $F'_\sharp \mu' = \mathrm{Unif}([0,1])$, and the second inequality uses that $G^{-1}$ is monotonic. If $F'$ is discontinuous, one should replace it with the kernel $\tilde{F}'$ which coincides with $F'$ where continuous and, at any point $x$ where there is a jump from $p_1$ to $p_2$, satisfies $\tilde{F}'_\sharp \delta_x = \mathrm{Unif}([p_1, p_2])$. By this choice, we have $\tilde{F}'_\sharp \mu = \mathrm{Unif}([0,1])$, and one can do the same for $F$ to obtain $\tilde{F}$ such that $\mathsf{W}_\infty(\tilde{F}_\sharp \mu, \tilde{F}'_\sharp \mu) \le \varepsilon$. At this point, we can derive the same bound as above. Now, writing $\Delta_i = G^{-1}((i+3)\varepsilon) - G^{-1}(i\varepsilon)$, we have

$$
\int_{\mathcal{X}} \left| G^{-1}(F(x)) - G^{-1}(F'(x)) \right|^p \mathrm{d}F'(x)
$$

$$
\le \varepsilon \sum_{i=-1}^{3M-2} \Delta_i^p
$$

$$
= \varepsilon \left( \sum_{i=0}^{M-1} \Delta_{3i-1}^p + \sum_{i=0}^{M-1} \Delta_{3i}^p + \sum_{i=0}^{M-1} \Delta_{3i+1}^p \right)
$$

$$
= \varepsilon D^p \left( \sum_{i=0}^{M-1} \left( \frac{\Delta_{3i-1}}{D} \right)^p + \sum_{i=0}^{M-1} \left( \frac{\Delta_{3i}}{D} \right)^p + \sum_{i=0}^{M-1} \left( \frac{\Delta_{3i+1}}{D} \right)^p \right)
$$

$$
\le \varepsilon D^p \left( \left( \sum_{i=0}^{M-1} \frac{\Delta_{3i-1}}{D} \right)^p + \left( \sum_{i=0}^{M-1} \frac{\Delta_{3i}}{D} \right)^p + \left( \sum_{i=0}^{M-1} \frac{\Delta_{3i+1}}{D} \right)^p \right)
$$

$$
= O(D\varepsilon^{1/p})^p.
$$

Thus, we have $\mathcal{E}_p(G^{-1} \circ F; \mu', \nu) \lesssim \| G^{-1} \circ F - G^{-1} \circ F' \|_{L^p(\mu')} \le D\varepsilon^{1/p}$, as desired. $\square$

Together, the three results stated above yield our desired risk bound.

**Proposition 2.** *Let $X_1, \ldots, X_n \overset{i.i.d.}{\sim} \mu \in \mathcal{P}(\mathbb{R})$ and $Y_1, \ldots, Y_n \overset{i.i.d.}{\sim} \nu \in \mathcal{P}([0,1])$. Then the estimator $\hat{\kappa}_n$ which, given $\hat{\mu}_n$ and $\hat{\nu}_n$, returns the optimal kernel for $\mathsf{W}_p(\hat{\mu}_n, \hat{\nu}_n)$ given by Lemma 12, achieves risk $\mathbb{E}[\mathcal{E}_p(\hat{\kappa}_n; \mu, \nu)] \lesssim n^{-1/(2p)}$.*

*Proof.* By Fact 2, we have that $\mathbb{E}[\| \mu - \hat{\mu}_n \|_{\mathrm{KS}}] \le n^{-1/2}$. Consequently, we bound

$$
\mathcal{E}_p(\hat{\kappa}_n; \mu, \nu) \le \mathcal{E}_p(\hat{\kappa}_n; \mu, \hat{\nu}_n) + \mathsf{W}_p(\nu, \hat{\nu}_n)
$$

$$
\le \| \mu - \hat{\mu}_n \|_{\mathrm{KS}}^{1/p} + \mathsf{W}_p(\nu, \hat{\nu}_n).
$$

Taking expectations and applying Fact 2 and Lemma 1 gives the desired rate. $\square$

Unfortunately, we are unaware of any multivariate extension of the KS distance that obeys a useful comparison inequality with $\mathsf{W}_p$ (like Fact 11) while maintaining strong empirical convergence guarantees (like Fact 2), inhibiting the further development of this approach.

# C  Additional Details for Section 4

We note that the minimax lower bounds in Corollaries 1 and 2 follow by combining the reduction to distribution estimation from Appendix B.2 with existing lower bounds for distribution estimation under $\mathsf{W}_p$ from Singh and Póczos [2018] and Weed and Berthet [2019], respectively.

# D  Proofs for Section 5

We first recall some basic facts used throughout.

**Fact 3** (TV contraction under Markov kernels). *For $\mu, \nu \in \mathcal{P}(\mathcal{X})$ and kernel $\kappa \in \mathcal{K}(\mathcal{X}, \mathcal{Y})$, we have $\| \kappa_\sharp \mu - \kappa_\sharp \nu \|_{\mathrm{TV}} \le \| \mu - \nu \|_{\mathrm{TV}}$.*

This follows by the data processing inequality.

**Fact 4** ($W_p$ contraction under convolution). *For $\mu, \nu, \alpha \in \mathcal{P}(\mathcal{X})$, we have $W_p(\mu * \alpha, \nu * \alpha) \leq W_p(\mu, \nu)$, where $*$ denotes convolution between probability measures.*

This follows by considering the couplings $(X + Z, Y + Z')$ of $\mu * \alpha$ and $\nu * \alpha$ which set $Z = Z'$.

**Fact 5** (TV discrete empirical convergence). *For a finite set $S$ with $|S| = k$, any distribution $\mu \in \Delta(S)$ exhibits empirical convergence in TV at rate $\mathbb{E}[\|\hat{\mu}_n - \mu\|_{\mathrm{TV}}] \lesssim \sqrt{k/n}$.*

To simplify discussion of our corruption model, we employ the $\varepsilon$-*outlier-robust $p$-Wasserstein distance*

$$W_p^\varepsilon(\mu, \nu) := \min_{\substack{\mu' \in \mathcal{P}(\mathbb{R}^d) \\ \|\mu' - \mu\|_{\mathrm{TV}} \leq \varepsilon}} W_p(\mu', \nu) = \min_{\substack{\nu' \in \mathcal{P}(\mathbb{R}^d) \\ \|\nu' - \nu\|_{\mathrm{TV}} \leq \varepsilon}} W_p(\mu, \nu'). \tag{9}$$

The second equality follows from the observation that, if $\mathbb{E}[\|X' - Y\|^p] \leq c$ and $X = X'$ with probability at least $1 - \varepsilon$, then the random variable $Y' = Y\mathbb{1}\{X = X'\} + X\mathbb{1}\{X \neq X'\}$ satisfies $\mathbb{E}[\|X - Y'\|^p] \leq c$. See Nietert et al. [2023a] for a thorough examination of $W_p^\varepsilon$ in the context of robust statistics. Under the setting of Section 5, our corruption model can be equivalently stated as follows: given the standard empirical measures $\hat{\mu}_n \in \mathcal{P}(\mathcal{X})$ and $\hat{\nu}_n \in \mathcal{P}(\mathcal{Y})$, we observe corrupted versions $\tilde{\mu}_n \in \mathcal{P}(\mathcal{X})$ and $\tilde{\mu}_n \in \mathcal{P}(\mathcal{Y})$ such that $W_p^\varepsilon(\tilde{\mu}_n, \hat{\mu}_n) \vee W_p^\varepsilon(\tilde{\nu}_n, \hat{\nu}_n) \leq \rho$.

For this setting, we handle sampling error using the following lemma, which mirrors Lemma 9.

**Lemma 13** (Prop. 2 of Goldfeld et al., 2020). *Fix $\sigma > 0$ and 1-sub-Gaussian $\mu \in \mathcal{P}(\mathbb{R}^d)$. Then, the $n$-sample empirical measure $\hat{\mu}_n$ satisfies $\mathbb{E}\left[\|N_\sharp^\sigma(\mu - \hat{\mu}_n)\|_{\mathrm{TV}}\right] \leq \sqrt{3^d(1 \vee \sigma^{-d})/n}$.*

In order to apply our $W_p$ stability result, Lemma 4, we use that any kernel become continuous if one first applies Gaussian convolution.

**Lemma 14.** *Fix $\bar{\kappa} \in \mathcal{K}(\mathcal{X}, \mathcal{Y})$, $\sigma > 0$, and let $\kappa = \bar{\kappa} \circ N^\sigma$. Then, for all $x, x' \in \mathcal{X}$, we have $W_p((\kappa_x, \kappa_{x'}) \leq \mathrm{diam}(\mathcal{Y})[\|x - x'\|/(2\sigma)]^{1/p}$.*

*Proof.* We simply compute

$$\begin{aligned} W_p(\kappa_x, \kappa_{x'}) &\leq \mathrm{diam}(\mathcal{Y})\|\kappa_\sharp(N_x^\sigma - N_{x'}^\sigma)\|_{\mathrm{TV}}^{1/p} && \text{(Fact 1)} \\ &\leq \mathrm{diam}(\mathcal{Y})\|N_x^\sigma - N_{x'}^\sigma\|_{\mathrm{TV}}^{1/p} && \text{(data processing ineq.)} \\ &\leq \mathrm{diam}(\mathcal{Y})\|\mathcal{N}(x, \sigma^2 I_d) - \mathcal{N}(x', \sigma^2 I_d)\|_{\mathrm{TV}}^{1/p} \\ &\leq \mathrm{diam}(\mathcal{Y})\|x - x'\|^{1/p}(2\sigma)^{-1/p}, \end{aligned}$$

where the final inequality follows by the closed form of KL divergence between Gaussians, combined with Pinsker's inequality. $\qquad\square$

We split the proof of Theorem 4 into the upper bound (Appendix D.1) and lower bound (Appendix D.2).

### D.1 Proof of Theorem 4 (Upper Bound)

To start, we decompose $N^\sigma = N^{\sigma_1 + \sigma_2} = N^{\sigma_1} \circ N^{\sigma_2}$, for $\sigma_1, \sigma_2$ to be tuned later. By our corruption model, there exists an intermediate measure $\mu_n' \in \mathcal{P}(\mathbb{R}^d)$ such that $\|\hat{\mu}_n - \mu_n'\|_{\mathrm{TV}} \leq \varepsilon$ and $W_p(\mu_n', \tilde{\mu}_n) \leq \rho$. By Facts 3 and 4, these bounds are preserved under convolution, so $\|N_\sharp^{\sigma_1}(\hat{\mu}_n - \mu_n')\|_{\mathrm{TV}} \leq \varepsilon$ and $W_p(N_\sharp^{\sigma_1}\mu_n', N_\sharp^{\sigma_1}\tilde{\mu}_n) \leq \rho$. By the TV triangle inequality, we have $\|N_\sharp^{\sigma_1}(\mu - \mu_n')\|_{\mathrm{TV}} \leq \tau := \varepsilon + \|N_\sharp^{\sigma_1}(\mu - \hat{\mu}_n)\|_{\mathrm{TV}}$. We conclude that $W_p^\tau(N_\sharp^{\sigma_1}\tilde{\mu}_n, N_\sharp^{\sigma_1}\mu) \leq \rho$. By the symmetric nature of $W_p^\tau$, there must also exist $\alpha \in \mathcal{P}(\mathbb{R}^d)$ such that $W_p(N_\sharp^{\sigma_1}\mu, \alpha) \leq \rho$ and $\|\alpha - N_\sharp^{\sigma_1}\tilde{\mu}_n\|_{\mathrm{TV}} \leq \tau$.

Now set $\bar{\kappa} = \kappa_p^\star[N_\sharp^\sigma \tilde{\mu}_n \to \tilde{\nu}_n]$, so that $\mathcal{E}_p(\bar{\kappa}; N_\sharp^\sigma \tilde{\mu}_n, \tilde{\nu}_n) = 0$. Using this, the TV bound above, and the fact that $N_\sharp^\sigma \tilde{\mu}_n = N_\sharp^{\sigma_2}(N_\sharp^{\sigma_1}\tilde{\mu}_n)$, we have $\mathcal{E}_p(\bar{\kappa}; N_\sharp^{\sigma_2}\alpha, \tilde{\nu}_n) \lesssim \mathrm{diam}(\mathcal{Y})\tau^{1/p} \leq \sqrt{d}\tau^{1/p}$. Applying Lemma 6, this gives

$$\mathcal{E}_p(\bar{\kappa} \circ N^{\sigma_2}; \alpha, \tilde{\nu}_n) \lesssim \sqrt{d}\tau^{\frac{1}{p}} + \mathbb{E}_{Z \sim \mathcal{N}(0, \sigma_2^2 I_d)}[\|Z\|^p]^{\frac{1}{p}} \lesssim \sqrt{d}\tau^{\frac{1}{p}} + \sqrt{d + p}\,\sigma_2.$$

Consequently, by Lemma 4 and Lemma 14, we have that

$$\mathcal{E}_p(\bar{\kappa} \circ N^{\sigma_2}; N_\sharp^{\sigma_1} \mu, \tilde{\nu}_n) \lesssim \mathcal{E}_p(\bar{\kappa} \circ N^{\sigma_2}; \alpha, \tilde{\nu}_n) + \rho + (\sqrt{d}/\sigma_2)^{1/p} \rho^{1/p}$$

$$\lesssim \mathcal{E}_p(\bar{\kappa} \circ N^{\sigma_2}; \alpha, \tilde{\nu}_n) + \rho + (\sqrt{d}/\sigma_2)^{1/p} \rho^{1/p}$$

$$\leq \sqrt{d} \tau^{\frac{1}{p}} + \rho + (\sqrt{d}/\sigma_2)^{1/p} \rho^{1/p} + \sqrt{d+p}\, \sigma_2.$$

Tuning $\sigma_2$ gives

$$\mathcal{E}_p(\bar{\kappa} \circ N^{\sigma_2}; N_\sharp^{\sigma_1} \mu, \tilde{\nu}_n) \lesssim \sqrt{d} \tau^{\frac{1}{p}} + \sqrt{d} \rho^{\frac{1}{p+1}} + \rho.$$

Apply Lemma 6 once more, we bound

$$\mathcal{E}_p(\bar{\kappa} \circ N^{\sigma}; \mu, \tilde{\nu}_n) \lesssim \sqrt{d} \tau^{\frac{1}{p}} + \sqrt{d} \rho^{\frac{1}{p+1}} + \rho + \sqrt{d+p}\, \sigma_1$$

$$\lesssim \sqrt{d} \varepsilon^{\frac{1}{p}} + \sqrt{d} \rho^{\frac{1}{p+1}} + \rho + \sqrt{d+p}\, \sigma_1 + \sqrt{d}\, \|N_\sharp^{\sigma_1}(\mu - \hat{\mu}_n)\|_{\mathrm{TV}}^{1/p}$$

Taking expectations and applying Lemma 13 yields

$$\mathbb{E}[\mathcal{E}_p(\bar{\kappa} \circ N^{\sigma}; \mu, \tilde{\nu}_n)] \lesssim \sqrt{d} \varepsilon^{\frac{1}{p}} + \sqrt{d} \rho^{\frac{1}{p+1}} + \rho + \sqrt{d+p}\, \sigma_1 + \mathbb{E}[\|N_\sharp^{\sigma_1}(\mu - \hat{\mu}_n)\|_{\mathrm{TV}}]^{\frac{1}{p}}$$

$$\lesssim \sqrt{d} \varepsilon^{\frac{1}{p}} + \sqrt{d} \rho^{\frac{1}{p+1}} + \rho + \sqrt{d+p}\, \sigma_1 + \left( \frac{3^d (1 \vee \sigma^{-d})}{n} \right)^{\frac{1}{2p}}.$$

Tuning $\sigma_1$ then gives

$$\mathbb{E}[\mathcal{E}_p(\bar{\kappa} \circ N^{\sigma}; \mu, \tilde{\nu}_n)] \lesssim \sqrt{d} \varepsilon^{\frac{1}{p}} + \sqrt{d} \rho^{\frac{1}{p+1}} + \rho + O_{p,d}(n^{-\frac{1}{d+2p}}).$$

Finally, we note that $\mathsf{W}_p(\tilde{\nu}_n, \hat{\nu}_n) \leq \rho + \sqrt{d} \varepsilon^{1/p}$ due to the support bound. Thus, Lemma 3 gives

$$\mathbb{E}[\mathcal{E}_p(\bar{\kappa} \circ N^{\sigma}; \mu, \nu)] \leq \mathbb{E}[\mathcal{E}_p(\bar{\kappa} \circ N^{\sigma}; \mu, \tilde{\nu}_n) + \mathsf{W}_p(\tilde{\nu}_n, \hat{\nu}_n) + \mathsf{W}_p(\hat{\nu}_n, \nu)]$$

$$\leq \mathbb{E}[\mathcal{E}_p(\bar{\kappa} \circ N^{\sigma}; \mu, \tilde{\nu}_n)] + \rho + \sqrt{d} \varepsilon^{\frac{1}{p}} + \mathbb{E}[\mathsf{W}_p(\hat{\nu}_n, \nu)]$$

$$\lesssim \sqrt{d} \varepsilon^{\frac{1}{p}} + \sqrt{d} \rho^{\frac{1}{p+1}} + \rho + O_{p,d}(n^{-\frac{1}{d+2p}}),$$

as desired.

For the null estimator, let $\kappa^\star$ be an optimal kernel for the $\mathsf{W}_p(\mu, \nu)$ problem and bound

$$\mathcal{E}(\hat{\kappa}_{\mathrm{null}}; \mu, \nu) = \left[ \left( \int \|x\|^p \mathrm{d}\mu(x) \right)^{\frac{1}{p}} - \left( \iint \|y - x\|^p \mathrm{d}\kappa_x^\star(y) \mathrm{d}\mu(x) \right)^{\frac{1}{p}} \right]_+ + \mathsf{W}_p(\delta_0, \nu)$$

$$\leq \left[ \left( \iint \|y\|^p \mathrm{d}\kappa_x^\star(y) \mathrm{d}\mu(x) \right)^{\frac{1}{p}} \right]_+ + \mathsf{W}_p(\delta_0, \nu) \qquad \text{(Minkowski's inequality)}$$

$$\leq 2\sqrt{d}, \qquad\qquad\qquad\qquad\qquad\qquad (\mathcal{Y} \subseteq [0, 1]^d)$$

as desired.

### D.2 Proof of Theorem 4 (Lower Bound)

Since $\sqrt{d} \varepsilon^{1/p}$ and $n^{-1/(d \vee 2p)}$ are less than $\sqrt{d}$, it suffices to prove a lower bound of $\sqrt{d} \varepsilon^{1/p} + d^{1/4} \rho^{1/2} \wedge \sqrt{d} + n^{-1/(d \vee 2p)}$. We inherit the $n^{-1/(d \vee 2p)}$ sampling error term of the lower bound from the Dirac mass construction described in Appendix B.2. For the remaining terms, we prove lower bounds which hold even in the infinite-sample population limit, and even when only the source measure is corrupted. Here, an estimator can be viewed as a map $\hat{\kappa}$ from $\mathcal{P}(\mathcal{X}) \times \mathcal{P}(\mathcal{Y}) \to \mathcal{K}(\mathcal{X}, \mathcal{Y})$, mapping the corrupted source measure $\mu$, guaranteed to satisfy $\mathsf{W}_p^\varepsilon(\tilde{\mu}, \mu) \leq \rho$, and the clean target measure $\nu$ to a kernel estimate $\hat{\kappa}[\tilde{\mu}, \nu]$. For $\mathcal{X} = \mathbb{B}^d$ (which, in particular, forces each $\mu \in \mathcal{P}(\mathcal{X})$ to be 1-sub-Gaussian) and $\mathcal{Y} = [-1, 1]^d$, we prove that

$$\sup_{\substack{\mu \in \mathcal{P}(\mathcal{X}) \\ \nu \in \mathcal{P}(\mathcal{Y})}} \sup_{\substack{\tilde{\mu} \in \mathcal{P}(\mathcal{X}) \\ \mathsf{W}_p^\varepsilon(\tilde{\mu}, \mu) \leq \rho}} \mathcal{E}_p(\hat{\kappa}[\tilde{\mu}, \nu]; \mu, \nu) \gtrsim \sqrt{d} \varepsilon^{\frac{1}{p}} + \rho^{\frac{1}{2}} d^{\frac{1}{4}} \wedge \sqrt{d}.$$

The choice of $\mathcal{Y} = [-1, 1]^d$ rather than $[0, 1]^d$ is solely to simplify notation in one of our constructions and can be reverted without loss. Finally, it suffices to lower bound the supremum by $\sqrt{d} \varepsilon^{1/p}$ when $\rho = 0$ and $\sqrt{d\rho} \wedge \sqrt{d}$ when $\varepsilon = 0$, separately, which we do presently.

**TV lower bound.** Fix target measure $\nu = (1 - \varepsilon)\delta_0 + \varepsilon\delta_y$, where $y = (1, \dots, 1) \in \mathbb{R}^d$. Consider the candidate clean measures $\mu_1 = \nu$ and $\mu_2 = \delta_0$. Because they are within TV distance $\varepsilon$, the observation $\tilde{\mu} = \nu$ is compatible with both candidates. Abbreviating $\kappa = \hat{\kappa}[\tilde{\mu}, \nu]$, we have

$$
\begin{aligned}
\mathcal{E}_p(\kappa; \mu_1, \nu) + \mathcal{E}_p(\kappa; \mu_2, \nu) &\geq \left[ \left( \iint \|y - x\|^p d\kappa_x(y) \mu_1(x) \right)^{\frac{1}{p}} - \mathsf{W}_p(\mu_1, \nu) \right]_+ + \mathsf{W}_p(\kappa_\sharp \mu_2, \nu) \\
&= \left( \iint \|y - x\|^p d\kappa_x(y) \nu(x) \right)^{\frac{1}{p}} + \mathsf{W}_p(\kappa_\sharp \delta_0, \nu) \\
&\geq (1 - \varepsilon)^{\frac{1}{p}} \left( \int \|y\|^p d\kappa_0(y) \right)^{\frac{1}{p}} + \mathsf{W}_p(\kappa_\sharp \delta_0, \nu) \\
&\geq (1 - \varepsilon)^{\frac{1}{p}} \mathsf{W}_p(\kappa_\sharp \delta_0, \delta_0) + (1 - \varepsilon)^{\frac{1}{p}} \mathsf{W}_p(\kappa_\sharp \delta_0, \nu) \\
&\geq (1 - \varepsilon)^{\frac{1}{p}} \mathsf{W}_p(\delta_0, \nu) \\
&\geq (1 - \varepsilon)^{\frac{1}{p}} \varepsilon^{\frac{1}{p}} \sqrt{d} \\
&\geq \frac{1}{2} \varepsilon^{\frac{1}{p}} \sqrt{d}.
\end{aligned}
$$

Thus, we must have $\mathcal{E}_p(\kappa; \mu_1, \nu) \vee \mathcal{E}_p(\kappa; \mu_2, \nu) \gtrsim \sqrt{d} \varepsilon^{1/p}$, as desired.

**$\mathsf{W}_p$ lower bound.** For the remaining bound, we first argue that, for any kernel $\kappa$, its performance for the $\mathsf{W}_p(\mu, \nu)$ problem cannot suffer to much if we compose it with the Euclidean projection onto $\mathrm{supp}(\nu)$, denoted by $\mathrm{proj}_{\mathrm{supp}(\nu)}$.

**Lemma 15.** *For $\mu \in \mathcal{P}(\mathcal{X})$, $\nu \in \mathcal{P}(\mathcal{Y})$, and $\kappa \in \mathcal{K}(\mathcal{X}, \mathcal{Y})$, we have*

$$
\mathcal{E}_p(\mathrm{proj}_{\mathrm{supp}(\nu)} \circ \kappa; \mu, \nu) \leq 4\mathcal{E}_p(\kappa; \mu, \nu).
$$

*Proof.* Write $f = \mathrm{proj}_{\mathrm{supp}(\nu)}$ and $\varepsilon = \mathcal{E}_p(\kappa; \mu, \nu)$. Fix a coupling $X, Y, Z$ such that $(X, Z) \sim (\mathrm{Id}, \kappa)_\sharp \mu$, $Y \sim \nu$, and $\mathbb{E}[\|Z - Y\|^p] = \mathsf{W}_p(\kappa_\sharp \mu, \nu)^p \leq \varepsilon^p$. Taking $Z' = f(Z)$, we then bound

$$
\begin{aligned}
\mathbb{E}[\|X - Z'\|^p]^{1/p} &\leq \mathbb{E}[\|X - Z\|^p]^{1/p} + \mathbb{E}[\|Z - Z'\|^p]^{1/p} \\
&= \mathbb{E}[\|X - Z\|^p]^{1/p} + \mathbb{E}[\|Z - f(Z)\|^p]^{1/p} \\
&\leq \mathsf{W}_p(\mu, \nu) + \varepsilon + \mathbb{E}[\|Z - Y\|^p]^{1/p} \\
&\leq \mathsf{W}_p(\mu, \nu) + 2\varepsilon.
\end{aligned}
$$

Similarly, we have

$$
\begin{aligned}
\mathsf{W}_p(\kappa'_\sharp \mu, \nu) &\leq \mathbb{E}[\|Z' - Y\|^p]^{1/p} \\
&\leq \mathbb{E}[\|Z - Y\|^p]^{1/p} + \mathbb{E}[\|Z - Z'\|^p]^{1/p} \\
&\leq \varepsilon + \varepsilon = 2\varepsilon.
\end{aligned}
$$

Thus, the sum of these two errors is at most $4\varepsilon$, as desired. $\square$

Now, fix target measure $\nu = \frac{1}{2}\delta_{-y} + \frac{1}{2}\delta_y$, where $y = (1, \dots, 1)$, and take $c \in [0, 1]$ to be tuned later. Then, for each $0 \leq t \leq 1/2$, define measure $\mu_t = (1/2 - t)\delta_{-cy} + (1/2 + t)\delta_{+cy}$. Now, fix any kernel $\kappa \in \mathcal{K}(\mathcal{X}, \{\pm y\})$, where the codomain restriction is without loss of generality due to Lemma 15. Note that its performance on each $\mu_t$ is determined by the two-point distributions

$\kappa_\pm := \kappa_{\pm cy} = (1 - \alpha_\pm)\delta_{-cy} + \alpha_\pm \delta_{cy}$. In particular, for $0 \le t < 1/2$, we compute

$$
\begin{aligned}
\mathsf{W}_p(\mu_t, \nu)^p &= \left(\tfrac{1}{2} - t\right)(1 - c)^p \|y\|^p + t(1 + c)^p \|y\|^p + \tfrac{1}{2}(1 - c)^p \|y\|^p \\
&= d^{\frac{p}{2}}\left[(1 - t)(1 - c)^p + t(1 + c)^p\right], \\
\mathsf{W}_p(\kappa_\sharp \mu_t, \nu)^p &= \mathsf{W}_p\left(\left(\tfrac{1}{2} - t\right)\kappa_- + \left(\tfrac{1}{2} + t\right)\kappa_+, \nu\right) \\
&= \|y - (-y)\|^p \cdot \mathsf{W}_p\left(\left(\tfrac{1}{2} - t\right)\mathrm{Ber}(\alpha_-) + \left(\tfrac{1}{2} + t\right)\mathrm{Ber}(\alpha_+), \mathrm{Ber}\left(\tfrac{1}{2}\right)\right)^p \\
&= (2d)^{\frac{p}{2}} \mathsf{W}_p\left(\mathrm{Ber}\left(\left(\tfrac{1}{2} - t\right)\alpha_- + \left(\tfrac{1}{2} + t\right)\alpha_+\right), \mathrm{Ber}\left(\tfrac{1}{2}\right)\right)^p \\
&= (2d)^{\frac{p}{2}} \left|\left(\tfrac{1}{2} - t\right)\alpha_- + \left(\tfrac{1}{2} + t\right)\alpha_+ - \tfrac{1}{2}\right|, \\
\mathsf{W}_p(\delta_{cy}, \kappa_+)^p &= \alpha_+(1 - c)^p \|y\|^p + (1 - \alpha_+)(1 + c)^p \|y\|^p \\
&= d^{\frac{p}{2}}\left(\alpha_+(1 - c)^p + (1 - \alpha_+)(1 + c)^p\right), \\
\mathsf{W}_p(\delta_{-cy}, \kappa_-)^p &= \alpha_-(1 + c)^p \|y\|^p + (1 - \alpha_-)(1 - c)^p \|y\|^p \\
&= d^{\frac{p}{2}}\left(\alpha_-(1 + c)^p + (1 - \alpha_+)(1 - c)^p\right), \\
\iint \|y - x\|^p \mathrm{d}\kappa_x(y)\mathrm{d}\mu_t(x) &= \left(\tfrac{1}{2} - t\right)\mathsf{W}_p(\delta_{-cy}, \kappa_-)^p + \left(\tfrac{1}{2} + t\right)\mathsf{W}_p(\delta_{cy}, \kappa_+)^p \\
&= d^{\frac{p}{2}}\Big[\left(\tfrac{1}{2} - t\right)\left(\alpha_-(1 + c)^p + (1 - \alpha_+)(1 - c)^p\right) \\
&\qquad + \left(\tfrac{1}{2} + t\right)\left(\alpha_+(1 - c)^p + (1 - \alpha_+)(1 + c)^p\right)\Big].
\end{aligned}
$$

Writing $\Delta = \alpha_- - \alpha_+$, we next bound

$$
\begin{aligned}
&\mathsf{W}_p(\kappa_\sharp \mu_t, \nu) + \mathsf{W}_p(\kappa_\sharp \mu_0, \nu) \\
&= \sqrt{2d}\left|\left(\tfrac{1}{2} - t\right)\alpha_- + \left(\tfrac{1}{2} + t\right)\alpha_+ - \tfrac{1}{2}\right|^{\frac{1}{p}} + \sqrt{2d}\left|\tfrac{1}{2}\alpha_- + \tfrac{1}{2}\alpha_+ - \tfrac{1}{2}\right|^{\frac{1}{p}} \\
&\ge \sqrt{2d}\, t^{\frac{1}{p}}|\alpha_+ - \alpha_-|^{\frac{1}{p}} \qquad\qquad\qquad\qquad\qquad \text{(subadditivity of } a \mapsto a^{1/p}) \\
&= \sqrt{2d}\, t^{\frac{1}{p}}|\Delta|^{\frac{1}{p}}
\end{aligned}
$$

and we simplify

$$
\begin{aligned}
\iint \|y - x\|^p \mathrm{d}\kappa_x(y)\mathrm{d}\mu_0(x) &= d^{\frac{p}{2}}\left(\frac{1 + \alpha_- - \alpha_+}{2}(1 + c)^p + \frac{1 - \alpha_- + \alpha_+}{2}(1 - c)^p\right) \\
&= d^{\frac{p}{2}}\left(\frac{1 + \Delta}{2}(1 + c)^p + \frac{1 - \Delta}{2}(1 - c)^p\right) \\
\mathsf{W}_p(\mu_0, \nu) &= \sqrt{d}\,(1 - c).
\end{aligned}
$$

Thus, we further bound

$$
\begin{aligned}
&\left[\left(\iint \|y - x\|^p \mathrm{d}\kappa_x(y)\mathrm{d}\mu_0(x)\right)^{\frac{1}{p}} - \mathsf{W}_p(\mu_0, \nu)\right]_+ \\
&= \sqrt{d}\left[\left(\frac{1 + \Delta}{2}(1 + c)^p + \frac{1 - \Delta}{2}(1 - c)^p\right)^{\frac{1}{p}} - 1 + c\right]_+ \\
&= \sqrt{d}\left[\left(\frac{1 + \Delta}{2}(1 + c)^p + \frac{1 - \Delta}{2}(1 - c)^p\right)^{\frac{1}{p}} - 1 + c\right].
\end{aligned}
$$

**Algorithm 1:** Randomized Rounding for Efficient OT Kernel Estimation

---

**Input:** $n$ corrupted source points $S \subseteq \mathbb{R}^d$ and target points $T \subseteq [0,1]^d$, budgets $\rho \geq 0$, $\varepsilon \in [0,1]$

1: $m \leftarrow n^2$, $\tau \leftarrow n^{-1/(d+2)}$, $\sigma \leftarrow 3^{d/(2+d)}(nd)^{-1/(d+2)} + \rho^{1/2}d^{-1/4}$
2: $S' \leftarrow \{\mathrm{proj}_S(X_i' + Z_i)\}_{i=1}^m$, where each $X_i' \sim S$ and $Z_i \sim \mathcal{N}_\sigma$ are sampled independently
3: Compute kernel $\bar\kappa \in \mathcal{K}(S',T)$ s.t. $\bar\kappa_\sharp \mathrm{Unif}(S') = \mathrm{Unif}(T)$

$$\frac{1}{m}\sum_{x\in S'}\int \|x - y\|\mathrm{d}\bar\kappa(y|x) \leq \mathsf{W}_1(S',T) + \tau$$

4: Return $\hat\kappa \in \mathcal{K}(\mathbb{R}^d,T)$ defined by $\hat\kappa = \bar\kappa \circ \mathrm{proj}_S \circ N^\sigma$

---

Combining, this gives

$$\mathcal{E}_p(\kappa;\mu_t,\nu) + \mathcal{E}_p(\kappa;\mu_0,\nu)$$

$$\geq \left[\sqrt{d}\left(\frac{1+\Delta}{2}(1+c)^p + \frac{1-\Delta}{2}(1-c)^p\right)^{\frac{1}{p}} - 1 + c + \sqrt{2d}\,t^{\frac{1}{p}}|\Delta|^{\frac{1}{p}}\right]$$

$$\geq \left[\sqrt{d}\left(\frac{1+\Delta}{2}(1+c) + \frac{1-\Delta}{2}(1-c)\right) - 1 + c + \sqrt{2d}\,t^{\frac{1}{p}}|\Delta|^{\frac{1}{p}}\right]$$

$$= \left[\sqrt{d}c(\Delta+1) + \sqrt{2d}\,t^{\frac{1}{p}}|\Delta|^{1/p}\right]$$

$$\geq \sqrt{d}\left[c(1-|\Delta|) + t^{1/p}|\Delta|\right]$$

$$\geq \sqrt{d}\min\{c/2, t^{1/p}/2\}$$

Now, supposing that $\rho < \sqrt{d}$, we can safely take $c = t^{1/p} = \rho^{1/2}d^{-1/4}/2$ while ensuring that $c \in [0,1]$ and $t \in [0,1/2]$, which were the only constraints on our construction. Otherwise, we take $c = t^{1/p} = 1/2$. In either case, we have $\mathsf{W}_p(\mu_0,\mu_t) = t^{1/p} \cdot 2c\sqrt{d} = (\rho \wedge \sqrt{d})/2 \leq \rho$. Thus, the observation $\tilde\mu = \mu_0$ is compatible with both $\mu = \mu_0$ and $\mu_t$ under our corruption model. This gives the desired minimax lower bound of $\Omega(\sqrt{d}c \wedge t^{1/p}) = \Omega(d^{1/4}\rho^{1/2} \wedge \sqrt{d})$.

### D.3 Efficient Computation

We now introduce Algorithm 1 to achieve efficient computation, focusing on $p = 1$ where we match the rate of Theorem 4. Here, we identify finite sets with their uniform distributions when convenient.

**Theorem 5** (Efficient implementation). *Under the setting of Section 5 with $p = 1$, the kernel $\hat\kappa$ returned by Algorithm 1 matches the risk bound of Theorem 4. Using an entropic OT solver for Step 3, Algorithm 1 runs in time $O((C_\infty + d)n^{2+o_d(1)})$, where $C_\infty = \max_{i,j}\|\tilde{X}_i - \tilde{Y}_j\|$. Moreover, $\hat\kappa$ can be evaluated (i.e., given $x \in \mathcal{X}$ we can sample $Y \sim \hat\kappa_x$) in time $O(nd)$.*

The proof below employs a similar analysis to that of Theorem 4, with multiple applications of Lemma 5 to account for various sampling errors along with TV contamination. We restrict to $p = 1$ due to the worsened scaling of Lemma 5 for $p > 1$.

*Proof.* Set $\alpha = N^\sigma_\sharp\tilde\mu_n$, $\beta = \mathrm{proj}_S\alpha$, and $\beta_m = \mathrm{Unif}(S')$. By construction, $S'$ is sampled i.i.d. from $\beta$, so Fact 5 gives that $\mathbb{E}[\|\beta - \beta_m\|_{\mathrm{TV}}] = \mathbb{E}[\mathbb{E}[\|\beta - \beta_m\|_{\mathrm{TV}}|S']] \lesssim \sqrt{n/m}$. Moreover,

$$\int \|\mathrm{proj}_S(x) - x\|\mathrm{d}\alpha(x) = \frac{1}{n}\sum_{x\in S}\int \|\mathrm{proj}_S(x+z) - x + z\|\mathrm{d}N^\sigma(z)$$

$$\leq \frac{1}{n}\sum_{x\in S}\int \|x - x + z\|\mathrm{d}N^\sigma(z) \qquad (x \in S)$$

$$= \int \|z\|\mathrm{d}N^\sigma(z)$$

$$\lesssim \sqrt{d}\,\sigma.$$

Now, we restate our guarantee for $\bar{\kappa}$; namely, we have:

$$\iint \|x - y\| \mathrm{d}\bar{\kappa}(y|x)\mathrm{d}\beta_m(x) \le \mathsf{W}_1(\beta_m, \tilde{\nu}_n) + \tau.$$

Thus, by Lemma 7, we have

$$\mathcal{E}_1(\bar{\kappa}; \beta, \tilde{\nu}_n) \lesssim \tau + \sqrt{d}\,\|\beta - \beta_m\|_{\mathrm{TV}},$$

and, applying Lemma 6, we obtain

$$\mathcal{E}_1(\bar{\kappa} \circ \mathrm{proj}_S; \alpha, \tilde{\nu}_n) \lesssim \tau + \sqrt{d}\,\|\beta - \beta_m\|_{\mathrm{TV}} + \sqrt{d}\,\sigma.$$

Now, write $\mu'_n \in \mathcal{P}(\mathbb{R}^d)$ for an intermediate measure such that $\|\mu'_n - \tilde{\mu}_n\|_{\mathrm{TV}} \le \varepsilon$ and $\mathsf{W}_p(\mu'_n, \hat{\mu}_n) \le \rho$. Noting that $\alpha = N^\sigma_\sharp \tilde{\mu}_n$, we have by Fact 3 that $\|\alpha - N^\sigma_\sharp \mu'_n\|_{\mathrm{TV}} \le \varepsilon$. Thus, Lemma 5 gives

$$\mathcal{E}_1(\bar{\kappa} \circ \mathrm{proj}_S; N^\sigma_\sharp \mu'_n, \tilde{\nu}_n) \lesssim \sqrt{d}\varepsilon + \tau + \sqrt{d}\,\|\beta - \beta_m\|_{\mathrm{TV}} + \sqrt{d}\,\sigma.$$

Applying Lemma 6 once more, we obtain

$$\mathcal{E}_1(\bar{\kappa} \circ \mathrm{proj}_S \circ N^{\sigma/2}; N^{\sigma/2}_\sharp \mu'_n, \tilde{\nu}_n) \lesssim \sqrt{d}\varepsilon + \tau + \sqrt{d}\,\|\beta - \beta_m\|_{\mathrm{TV}} + \sqrt{d}\,\sigma.$$

By Lemma 14, the fact that this latest kernel begins with the convolution $N^{\sigma/2}$ ensures that it is $O(\sqrt{d}\sigma^{-1})$-Lipschitz w.r.t. $\mathsf{W}_1$. Moreover, by Fact 4, we have $\mathsf{W}_1(N^{\sigma/2}_\sharp \mu'_n, N^{\sigma/2}_\sharp \hat{\mu}_n) \le \mathsf{W}_1(\mu'_n, \hat{\mu}_n) \le \rho$. Thus, Lemma 4 gives

$$\mathcal{E}_1(\bar{\kappa} \circ \mathrm{proj}_S \circ N^{\sigma/2}; N^{\sigma/2}_\sharp \hat{\mu}_n, \tilde{\nu}_n) \lesssim \sqrt{d}\varepsilon + \tau + \sqrt{d}\,\|\beta - \beta_m\|_{\mathrm{TV}} + \sqrt{d}\,\sigma + \rho + \frac{\sqrt{d}\,\rho}{\sigma}.$$

Next, we apply Lemma 5 and Lemma 6 to bound

$$\begin{aligned}
&\mathcal{E}_1(\bar{\kappa} \circ \mathrm{proj}_S \circ N^\sigma; \mu, \tilde{\nu}_n)\\
&\lesssim \mathcal{E}_1(\bar{\kappa} \circ \mathrm{proj}_S \circ N^{\sigma/2}; N^{\sigma/2}_\sharp \mu, \tilde{\nu}_n) + \sqrt{d}\,\sigma\\
&\lesssim \mathcal{E}_1(\bar{\kappa} \circ \mathrm{proj}_S \circ N^{\sigma/2}; N^{\sigma/2}_\sharp \hat{\mu}_n, \tilde{\nu}_n) + \sqrt{d}\,\sigma + \sqrt{d}\,\|N^{\sigma/2}(\mu - \hat{\mu}_n)\|_{\mathrm{TV}}\\
&\lesssim \sqrt{d}\,\varepsilon + \tau + \sqrt{d}\,\sigma + \rho + \frac{\sqrt{d}\,\rho}{\sigma} + \sqrt{d}\,\|\beta - \beta_m\|_{\mathrm{TV}} + \sqrt{d}\,\|N^{\sigma/2}(\mu - \hat{\mu}_n)\|_{\mathrm{TV}}.
\end{aligned}$$

Finally, we correct the target measure, using Lemma 3 to bound

$$\begin{aligned}
&\mathcal{E}_1(\bar{\kappa} \circ \mathrm{proj}_S \circ N^\sigma; \mu, \nu) \le \mathcal{E}_1(\bar{\kappa} \circ \mathrm{proj}_S \circ N^\sigma; \mu, \tilde{\nu}_n) + 2\mathsf{W}_p(\tilde{\nu}_n, \nu)\\
&\lesssim \sqrt{d}\,\varepsilon + \tau + \sqrt{d}\,\sigma + \rho + \frac{\sqrt{d}\,\rho}{\sigma} + \sqrt{d}\,\|\beta - \beta_m\|_{\mathrm{TV}} + \sqrt{d}\,\|N^{\sigma/2}(\mu - \hat{\mu}_n)\|_{\mathrm{TV}} + \mathsf{W}_p(\hat{\nu}_n, \nu)
\end{aligned}$$

Taking expectations, using our early bound on the first TV distance, and applying Lemma 13 for the second TV distance, and applying Lemma 1 for the Wasserstein distance, we obtain

$$\begin{aligned}
&\mathbb{E}[\mathcal{E}_1(\bar{\kappa} \circ \mathrm{proj}_S \circ N^\sigma; \mu, \nu)]\\
&\lesssim \sqrt{d}\,\varepsilon + \tau + \sqrt{d}\,\sigma + \rho + \frac{\sqrt{d}\,\rho}{\sigma} + \sqrt{\frac{dn}{m}} + \sqrt{d\,3^d(1 \vee \sigma^{-d})/n} + c_{p,d}n^{-\frac{1}{p \vee 2d}}\log^2 n
\end{aligned}$$

Our choice of $\sigma$, $m$, and $\tau$ ensure that the desired risk bound holds.

Computational complexity is dominated by the OT computation at Step 3. The source and target distributions are both supported on $n$ points, and we require accuracy $\tau = n^{-1/(d+2)}$. Computing the relevant cost matrix requires time $O(n^2 d)$. Using a state of the art OT solver based on entropic OT (e.g., Luo et al., 2023) gives a running time of $O(C_\infty n^2/\tau) = O(C_\infty n^{2+1/(d+2)})$, where $C_\infty$ is the largest distance between a point in $S$ and a point in $T$. Combining these two gives the first bound. Evaluation complexity is dominated by the projection step, which can be computed in a brute-force manner using $O(nd)$ time. $\qquad\square$

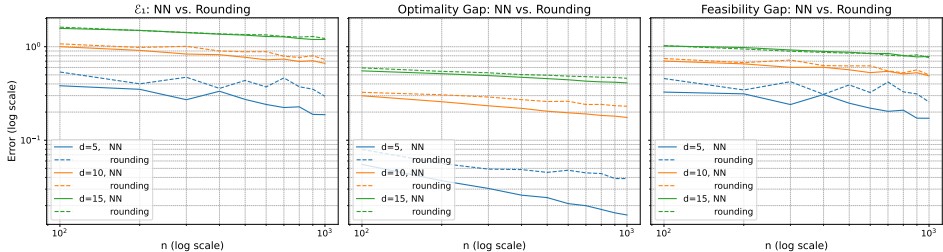

Figure 4: $\mathcal{E}_1$ (left), optimality gap (middle), and feasibility gap (right) performance of nearest-neighbor and rounding estimators for Setting A.

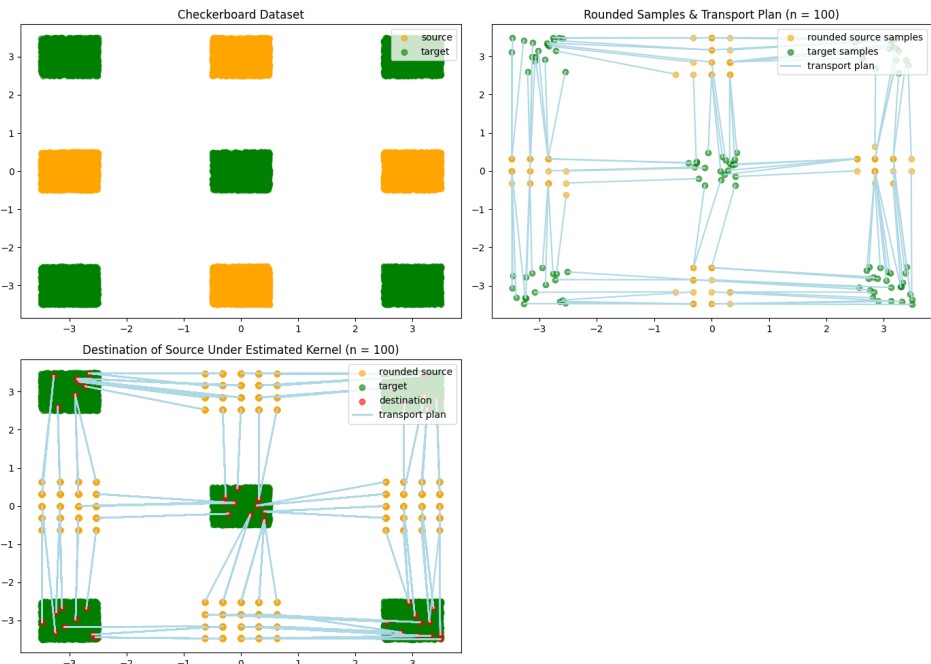

Figure 5: Visual depiction of rounding kernel estimator on checkerboard dataset.

## E  Additional Experiments

All code needed to reproduce our experiments and figures is available at `https://github.com/sbnietert/map-estimation`. Here, we include two additional experiments beyond those in the main body, one with higher dimensions and sample sizes, and one in dimension two, for visualization.

First, in Figure 3, we repeat the Setting A experiments from Section 6 (Figure 3, middle), but with $N = 10000$, dimensions $d \in \{5, 10, 15\}$, and sample sizes $n \in \{100, 200, \ldots, 1000\}$. To extend to these larger parameters, we reduced the number of iterations to $T = 5$ and omitted the bootstrapped error bars. Also, we include the decomposition of $\mathcal{E}_1$ into its optimality gap and feasibility gap components, the latter of which is measurably larger. As predicted by our analysis, our error rates worsen with dimension.

Finally, in Figure 5, we provide a visual depiction of the rounding estimator on a toy checkerboard dataset. In the top left, we present our source measure (orange) and target measure (green). For the top right plot, we sampled $n = 100$ source and target samples, rounded the source samples onto a regular grid with side length $\delta = n^{-1/(d+2)}$ (orange), and computed an OT plan (light blue) from the rounded source samples to the target samples (green). For the bottom left, we rounded the full source distribution onto the same grid (orange), route these according to the same OT plan (light blue), reaching a destination measure (red) that approximates the target distribution (green).

