# OpenReview forum: "Estimation of Stochastic Optimal Transport Maps"
_NeurIPS.cc/2025/Conference — NeurIPS 2025 poster_

### Official Review · Reviewer_RJR5 · 2025-06-30

**Clarity:** 3
**Significance:** 3
**Originality:** 3
**Rating:** 5
**Confidence:** 4

**Summary:**

This paper introduces a new measure of error for transition kernels $\kappa(y|x)$ used as stochastic transportation maps between $\nu$ and $\mu$, coined *Transportation Error* $\mathcal{E}_{p}(\kappa,\mu,\nu)$ which decomposes transportation cost into two terms:
- the *optimality gap* induces by the mapping $\kappa$ and the optimal $T^*$
- the *feasibility gap* that quantifies the failure to meet the pushforward constraint $\nu = T\sharp \mu$

This decomposition has several advantages:
- better models of errors under TV or Wasserstein perturbations
- "weaker" convergence than $L_p$ - which is strength for estimation from finite samples, like for Wasserstein cost
- stability to perturbations wrt $\nu$
- stability to perturbations wrt $\mu$
- "kernel decomposition" property to link errors of $\kappa_1$ and $\kappa_2$ with $\kappa_1\circ\kappa_2$

Finally, the authors propose an algorithm whose sample complexity approaches the optimal minimax rate. This algorithms relies on the convexity property of entropy-regularized OT, which yield a consistent estimator, well defined outside the support of the "train set".

**Questions:**

### Toy-ish experiments

I'd be curious to see how your estimator behaves beyond the N=2,000 points limit, which is very small by today's standards. Can you plot results for $N\geq 10^4$ points [you may reduce or remove bootstrap if computational budget is an issue]?

Also, other "challenging" OT problems are documented in literature, with discontinuous OT maps. For example, the ones in Fig 3 of Makkuva et al. Also, the "checkerboard" dataset, used in e.g. Vesseron et al. It could be interesting to see how your estimator behaves on these datasets.

[Makkuva et al] Makkuva, Ashok, Amirhossein Taghvaei, Sewoong Oh, and Jason Lee. "Optimal transport mapping via input convex neural networks." In International Conference on Machine Learning, pp. 6672-6681. PMLR, 2020.

I'd be willing to increase my score if these experiments are done.

### Extension to deep learning

Note that one of the paper's selling points is: "*Adopting Ep as our objective enables kernel estimation without regularity or even existence of a Brenier map*". Many folks in the deep learning community care about an objective which is:
- statistically consistent, so it can be estimated from a finite sample in minibatch setup (as you did)
- is differentiable so it can be used as a loss function in a deep learning context
e.g. with a kernel $\kappa$ represented as a neural network.

Can you discuss the use of $\mathcal{E}_p$ in this context?

**Ethical Concerns:**

["NO or VERY MINOR ethics concerns only"]

**Final Justification:**

I am satisfied by the answers of authors, in particular the efforts they put in comparing their work with the one of Uscida & Cuturi.

I increased my rating, upon the assumption that the additional experiments promised by the authors are present in the camera ready version.

**Limitations:**

Some limitations are discussed, including the "surprising" efficiency of the NN estimator.  However, these experiments are toy, so while the transportation error $\mathcal{E}_p$ is certainly a convincing theoretical tool, its practical relevance to "large scale" scenarios and realistic datasets still needs to be demonstrated. A lengthy discussion on this topic would be welcome.

**Paper Formatting Concerns:**

No concerns.

**Quality:**

3

**Strengths And Weaknesses:**

## Strengths

The problem is well-exposed and well-motivated. To the best of my knowledge, this approach is new.

The flow is clear, and as pedagogical as one can be on these topics (e.g. Figure 2).

From the paper: "Adopting $\mathcal{E}_{p}(\kappa,\mu,\nu)$ as our objective enables kernel estimation without regularity or even existence of a Brenier map" which may simplify assumptions or algorithms built upon it.

The algorithm proposed starts with a textbook application of the properties of entropy-regularized OT, based on the work of Genevay et al, and then apply "rounding" trick to achieve better rates.

When $p\geq 2$, using a Wasserstein distributionally robust optimization program, authors relaxed the frequent Lipschitz continuity assumption put on $T$ and $T^{-1}$.

Code is given.

## Weaknesses

### Missing related work

The proposed *Transportation Error* is closely related to the *Monge Gap* [Uscida & Cuturi] which performs a similar-looking decomposition between a *fitting term* $\Delta(T\sharp \nu, \mu)$ (playing the same role as the *feasibility gap*) and a *c-optimality* term which measures how different $T$ is from $T^*$ on the induced transportation cost. I acknowledge that their decomposition is not quite the same, nonetheless, the differences could be discussed and highlighted.

[Uscida & Cuturi] Uscidda, Théo, and Marco Cuturi. "The monge gap: A regularizer to learn all transport maps." In International Conference on Machine Learning, pp. 34709-34733. PMLR, 2023.

### Toy experiments

Please see the "questions" section.

## Recommendation

Overall, the paper is clear and introduces a novel tool to measure the optimality of transition kernels $\kappa$ used as stochastic transportation maps. These transportation maps are paramount to some modern generative models (e.g. GAN-like or flow matching). This paper introduces a new tool to evaluate their quality, with appealing theoretical properties, and a practical algorithm to evaluate it. I believe the NeurIPS community will find it useful.

---

> ### Author Rebuttal · Authors · 2025-07-31
>
> We thank the reviewer for their thoughtful feedback and clarifying questions.
>
> **Weakness, Monge gap literature:** Thank you for bringing up this work, which is indeed quite relevant. When specialized to $p=1$, the Monge gap paper [A] aims to learn a parameterized map $T$ minimizing the following:
> $$\mathcal{E}’\_1(T;P,Q) := \underbrace{\mathbb{E}\_P[\\|T(X) - X\\|] - W\_1(P,T\_\sharp P)}\_{\text{Monge gap}} + \Delta(T\_\sharp P,Q),$$
> where $\Delta$ is some statistical divergence. In practice, they substitute $P$ and $Q$ with their empirical versions and use entropic OT (EOT) distances to approximate both $W\_1$ and $\Delta$, achieving impressive empirical results. We emphasize that these approximations preclude meaningful statistical guarantees without accounting for implicit bias or other regularization, since an optimal map for the empirical problem can perform arbitrarily bad on the population problem.
>
> We can show that, when $\Delta = W\_1$, the Monge gap objective $\mathcal{E}’\_1$ and our $\mathcal{E}\_1$ coincide up to a constant multiplicative factor of 2. Qualitatively similar results hold for $p>1$, though there is no longer a precise equivalence. Thus, the approach in [A] is quite close to performing empirical risk minimization with our transportation error (up to the use of EOT that is necessary for large-scale data sets). We will include this derivation in a new remark for Section 2. In some sense, this shows that our approach already lends itself to map fitting in practice. Moreover, if we were to directly plug in our $\mathcal{E}\_p$ objective into the neural map estimation experiments from [A] (substituting OT with EOT for tractability), we have the benefit of only having to perform back propagation through one EOT problem instead of two. We are optimistic that this will lead to a performance boost in practice. In the updated manuscript, we will supplement our synthetic experiments, which validated our theory, with a large-scale experiment that compares our performance to the Monge gap approach when learning maps for single-cell genomic data (mirroring the setup of Section 6.4 in [A]). Zooming out, we feel that our framework provides a solid statistical foundation for the practical approach to map estimation established in [A], and we hope that our theoretical insights will boost further practical developments.
>
> Second, our rounding estimation approach can be applied in a black-box way on top of any existing map estimator, including a neural estimator. In the added experiments with single-cell genomic data, we will take the resulting neural estimators and see how their test performance varies if we add on this rounding step, for varied choices of rounding partition (namely, tuning the size of the partition cells). It will be interesting to see if there is extra performance to be eked out of existing methods via the regularizing effect of rounding.
>
> Third, we call the reviewer’s attention to our Remark 4, which draws parallels between our DRO procedure and existing Lipschitz regularization techniques.
>
> Finally, we remark that our current synthetic experiments were computationally bottlenecked by high-precision OT computations, which were necessary to validate our statistical guarantees. The Monge gap work sidesteps these by using EOT with a fixed precision, trading some accuracy for scalability. Another regularization approach which may be worth future exploration is that of sliced OT, where one works only with low dimensional projections of the data distributions. We will add a remark to the end of Section 2 on extensions of our framework to regularized OT problems, which are key for large-scale applications.
>
> **Question, toy experiments:** We agree that our initial synthetic benchmarks were somewhat limited. In addition to the proposals above, we are happy to include the recommended toy setting in the revised manuscript.
>
> **Question, deep learning extensions:** We hope that our response above re: Monge gap is satisfactory. In particular, we note that our proposal with the rounding estimator relies on our extension to stochastic maps and is rooted in deep learning (of course, empirical evaluation awaits). One could hope for a neural estimator that involves a more explicit parameterization of kernels, rather than an indirect approach via rounding. This is an interesting question for future research which we will include as an open direction in the updated conclusion. Certainly, well-crafted regularization would be essential, in order to prevent variance terms from exploding due to the very high-dimensional search space in this formulation.
>
> [A]: Uscidda & Cuturi. “The Monge Gap: A Regularizer to Learn All Transport Maps”. ICML 2023.

---

> > ### Comment · Reviewer_RJR5 · 2025-08-03
> >
> > Thank you for your answer. I find it satisfactory.
> >
> > I increased my score upon the assumption the experiments you promised are present in the camera ready version.

---

### Official Review · Reviewer_LQyn · 2025-07-01

**Clarity:** 2
**Significance:** 2
**Originality:** 3
**Rating:** 3
**Confidence:** 3

**Summary:**

The authors find optimal map $\pi_{\tau,n}$ in Entropic OT problem statement between data distributions and represent conditional kernel between data distributions through this map. Having armed with the kernel, they build a novel error metric $\epsilon_{p}$  for validation of Optimal Transport (OT) solvers  that demonstrates better performance than $L_{p}$ with increasing of number of samples. Also, they create rounding estimator to increase the efficiency of developed methodology, providing comparison with nearest-neighbour estimator in two settings.

**Questions:**

1) Why do you use this error metric as validation function and not use as a training objective?

2) Since barycentric projection is averaging of data, it seems that is poor representation for map. Is there opportunity to learn 2 neural networks, while the first approximates map the second learns conditional kernel?

3) Is your error unbiased? Could you show the proof of this fact, please

4) Why do you pay your attention on Entropic OT to get transport plan? Is there any ways to get transport plan to obtain map without barycentric projection?

5) If I am not mistaken, the theory fails if we remove sub-Gaussian requirement, yes?

6) I poorly understood sense of rounding-based estimators. Could you explain what is the main motivation of rounding-based estimator?


7) Could you provide high-dimensional experiments with MNIST or CIFAR-10 dataset, please?

**Ethical Concerns:**

["NO or VERY MINOR ethics concerns only"]

**Final Justification:**

Unfortunately, I remain my score unchanged. I really appreciate experiments, but I am sure that large-scale domain adaptation experiments would significantly improve the quality of this paper.

Sincerely

**Limitations:**

The method has the same limitations as Entropic OT (see weakness 1).

**Paper Formatting Concerns:**

I have no concerns.

**Quality:**

3

**Strengths And Weaknesses:**

**strengths**

1) The developed error does not require uniqueness or even existence of OT map.

2) The developed error $\epsilon_{1}$ demonstrates better performance than $L_{1}$

**weaknesses**

1) They solve Entropic OT to define conditional kernel in error metric. As far as known, Entropic OT has some drawbacks such as instability (exponents in  KL regularization),  sharpening of generated data (barycentric projection is averaging) and  tuning of hyper parameters.

2)  Although the proposed methodology is applicable for any $p\neq2$, the authors demonstrate experiments only for $p=1$. (see Fig.3)

3)  Since 10 is the highest dimension for the experiments (see Sec.6, line 334), there is no confidence that this methodology is scalable. There is no Image experiments with MNIST or CIFAR-10 dataset.

4) This methodology is applicable only for $\sigma^{2}$-sub Gaussian data distribution.(line 120)

---

> ### Author Rebuttal · Authors · 2025-07-31
>
> We thank the reviewer for their thoughtful feedback and clarifying questions.
>
> **Weakness 1:** “They solve Entropic OT to define conditional kernel in error metric. As far as known, Entropic OT has some drawbacks such as instability (exponents in KL regularization), sharpening of generated data (barycentric projection is averaging) and tuning of hyper parameters.”
> The reviewer is fair to point out some known drawbacks of EOT. We want to emphasize that EOT is not part of our error metric, and our guarantees for the entropic kernel estimator in Section 3 are strictly worse than those for the rounding estimator. Nonetheless, EOT remains widely used in practice, so we wanted to comment on its performance when applied to our estimation task. We also would like to clarify that our estimators do not use barycentric projection.
>
> **Weakness 2, p=1 for experiments:** We chose p=1 for the experiments to emphasize the application of our framework even when uniqueness of the OT map fails dramatically (when p=1, there are many optimal maps). However, this is not a contributing factor to computation - we will include an updated set of experiments for p=2 in the updated appendix. Indeed, OT solvers with high accuracy guarantees are mostly agnostic to the choice of cost. For the cases of p=1,2, there are also efficient neural solvers using input-Lipschitz and input-convex neural networks, respectively. (See further discussion in response to the next weakness.)
>
> **Weakness 3, scalability concerns:** The reviewer fairly points out that we do not use our methods in truly large scale settings. As a minor point, we note that our framework opens up the potential for analysis of future algorithms under far weaker assumptions than currently standard. But, more importantly, we are confident that our techniques may have immediate applications for neural map estimation with large-scale data, via a connection to the Monge gap discussed below.
> As noted by Reviewer RJR5, there is a promising approach for neural map estimation using the so-called “Monge gap” [A], which we will discuss in the updated related work. When specialized to $p=1$, this work aims to learn a parameterized map $T$ minimizing the following:
> $$\mathcal{E}’\_1(T;P,Q) := \underbrace{\mathbb{E}\_P[\\|T(X) - X\\|] - W\_1(P,T\_\sharp P)}\_{\text{Monge gap}} + \Delta(T\_\sharp P,Q),$$
> where $\Delta$ is some statistical divergence. In practice, they substitute P and Q with their empirical versions and use entropic OT (EOT) distances to approximate both $W\_1$ and $\Delta$, achieving impressive empirical results. We emphasize that these approximations preclude meaningful statistical guarantees without accounting for implicit bias or other regularization, since an optimal map for the empirical problem can perform arbitrarily bad on the population problem.
>
> We can show that, when $\Delta = W\_1$, the Monge gap objective $\mathcal{E}’_1$ and our $\mathcal{E}_1$ coincide up to a constant multiplicative factor of 2. Qualitatively similar results hold for $p>1$, though there is no longer a precise equivalence. Thus, the approach in [A] is quite close to performing empirical risk minimization with our transportation error (up to the use of EOT that is necessary for large-scale data sets). We will include this derivation in a new remark for Section 2. In some sense, this shows that our approach already lends itself to map fitting in practice. Moreover, if we were to directly plug in our $\mathcal{E}_p$ objective into the neural map estimation experiments from [A] (substituting OT with EOT for tractability), we have the benefit of only having to perform back propagation through one EOT problem instead of two. We are optimistic that this will lead to a performance boost in practice. In the updated manuscript, we will supplement our synthetic experiments, which validated our theory, with a large-scale experiment that compares our performance to the Monge gap approach when learning maps for single-cell genomic data (mirroring the setup of Section 6.4 in [A]). Zooming out, we feel that our framework provides a solid statistical foundation for the practical approach to map estimation established in [A], and we hope that our theoretical insights will boost further practical developments.
>
> Second, our rounding estimation approach can be applied in a black-box way on top of any existing map estimator, including a neural estimator. In the added experiments with single-cell genomic data, we will take the resulting neural estimators and see how their test performance varies if we add on this rounding step, for varied choices of rounding partition (namely, tuning the size of the partition cells). It will be interesting to see if there is extra performance to be eked out of existing methods via the regularizing effect of rounding.
>
> Third, we remark that our current synthetic experiments were computationally bottlenecked by high-precision OT computations, which were necessary to validate our statistical guarantees. The Monge gap work sidesteps these by using EOT with a fixed precision, trading some accuracy for scalability. Another regularization approach which may be worth future exploration is that of sliced OT, where one works only with low dimensional projections of the data distributions. Finally, one could also employ neural OT solvers, which compute W1 and W2 using neural parameterizations of Lipschitz and convex function classes, respectively. We will add a remark to the end of Section 2 on extensions of our framework to regularized/approximate OT problems, which are key for large-scale applications.
>
> Finally, we will add an appendix section which pushes the dimension and sample count as high as possible for our existing high-precision, synthetic experiments. We expect that they can be pushed a fair bit further if bootstrapping is omitted.
>
> **Weakness 4, sub-Gaussianity:** As noted in Theorem 2, we can relax the sub-Gaussianity assumption on $\mu$ by using a non-uniform partition for the rounding algorithm. One can also employ weaker tail bounds on $\nu$, but we expect that these will necessarily worsen the achievable risk (as remarked after Theorem 2). We note that the existing statistics literature generally required bounded support, a stronger assumption.
>
> **Q1, evaluation vs optimization objective:** This is certainly a natural question. Since we were primarily focused on statistical guarantees, we knew that performing empirical risk minimization with $\mathcal{E}\_p$ as our objective was unfortunately out of the question, since the optimal empirical map can perform arbitrarily poorly on the population measures. However, our approach in Section 4, using Wasserstein distributionally robust optimization, does use $\mathcal{E}\_p$ as an objective for robust optimization (and can be interpreted as a regularized version of ERM, see Remark 4). Finally, we hope that our promised large-scale experiments above, which compare to the Monge gap approach, will shed some empirical light on the merit of $\mathcal{E}\_p$ for neural map estimation.
>
> **Q2a, barycentric projection:** We agree that barycentric projection is problematic. This is a feature of some prior work, but we emphasize that none of our estimators use barycentric projection. We would appreciate if the reviewer could point out which part of our text caused this confusion (namely, the wrong impression that barycentric projection was employed in our error metric or estimators) and we will work to elucidate it for the final version.
>
> **Q2b, two neural networks:** We kindly ask that the reviewer clarify this suggestion (we are happy to discuss further during the discussion period).
>
> **Q3, unbiasedness:** It is unclear to us what a suitable notion of unbiasedness would be for this paper's setting (since we are not comparing the estimated map to a fixed optimal map $T^\star$). Does the reviewer have any particular definition in mind?
>
> **Q4, EOT & barycentric projection:** As mentioned above, we do not perform barycentric projection, and our best estimation guarantees do not use EOT.
>
> **Q5, sub-Gaussianity:** We point the reviewer to our response for Weakness 4.
>
> **Q6, intuition for rounding:** At a high-level, our rounding procedure allows us to extend a map learned from a discrete point set to the full source domain, while also serving to reduce variance. Recall that the rounding estimator bins the source points into bins of a tuned size, learns a map from the binned points to the (untouched) target points, and, given a new point, sends it to the destination of its bin under the learned map. The variance reduction intuition is as follows: at high spatial resolution, OT analysis is typically dominated by sampling variance terms, and binning serves to keep our analysis at a moderate spatial resolution. Of course, binning also introduces some bias. Balancing these factors was the key to our proof of Theorem 2. We include this approach because it achieves sharper statistical rates than the estimator using EOT.
>
> **Q7, large-scale experiments:** As mentioned above, the role of our current experiments was to validate our statistical rates, requiring high-precision OT computations that were a bottleneck to very high-dimensions / sample sizes. Please see our response to the scalability concern above for a new set of neural estimation experiments to be included in the revised manuscript. We hope that the single-cell genomics experiment is satisfactory as a large-scale example. This is a compelling option due to the existence of past results to benchmark against.
>
> **Limitations, EOT:** Please see previous responses on EOT.
>
> [A]: Uscidda & Cuturi. “The Monge Gap: A Regularizer to Learn All Transport Maps”. ICML 2023.

---

> > ### Comment · Reviewer_LQyn · 2025-08-07
> >
> > Dear authors
> >
> > I am exceedingly grateful for your response
> >
> > 1) I really appreciate that you provide experiments with single-cell genomics. However, one would like to look at performance of your method with unconditional generation on Celeba or FFHQ as well as high-dimensional translation female to male or winter to summer with 64 resolution at least. Since SOTA OT methods [1] translate data from one domain to another, one would like to look at performance of your approach.
> >
> > 2) If I am not mistaken your approach cannot generate arbitrary distribution on;y Sub-Gaussian, don't I?
> >
> > 3) I realized that you don't use EOT solution in your approach. If I am not mistaken is new paradigm to transfer data in OT manner from one domain to another, don't I?
> >
> > [1] - Korotin et al "Neural Optimal Transport"

---

> > > ### Author Response · Authors · 2025-08-08
> > >
> > > 1) Thank you for bringing this reference [1] to our attention - we will certainly include it in the updated related work. We want to emphasize that the contributions of [1] are primarily empirical, demonstrating the effectiveness of certain neural map estimators, whereas our primary contributions are a novel framework for statistical analysis and our accompanying estimation guarantees. We note that there is precedent for statistical map estimation results being published at top-tier ML venues (see, e.g., [A,B,C] below), where experiments are often omitted or limited to toy settings / validation of theory. We believe that our promised set of updated experiments, including neural estimators, exceeds the existing bar for such papers at this conference. Nonetheless, if the reviewing team feels that large-scale domain adaptation experiments would significantly improve the quality of this paper, we can apply our previously proposed neural estimator to this task with some benchmark image datasets, as suggested. However, we note that strong performance on such tasks often requires careful architecture choices and parameter tuning which we feel are slightly beyond the current scope.
> > >
> > > 2) Our error metric is well-defined without any assumptions on the input measures $\mu$ and $\nu$ (and is finite assuming only bounded $p$th moments). For quantitative finite-sample estimation rates, some assumptions are needed, but sub-Gaussianity is not a requirement for either $\mu$ or $\nu$ (although it does lead to a simpler presentation). As a simple example, in the setting of Section 4, Assumption 1 is satisfied with $\alpha = L = 1$ in cases where the target measure is a translation of the source measure. Then, Corollary 1 gives a guarantee of order $n^{-1/\max\\{d,2p\\}}$ under bounded $2p$th moments (even without sub-Gaussianity). For the setting of Section 3 (where our estimators are efficient), the sub-Gaussian assumption can also be relaxed, as alluded to in our response above. For a concrete example, suppose that both the source and target measure only have bounded $2p$th moments. Our current analysis for Theorem 2 already tolerates this weak tail bound for the source measure $\mu$. For the target measure $\nu$, we can show that an alternative choice of the trimming radius $R$ gives an error bound with rate $n^{-1/\max\\{2d,4p\\}}$ (square root of current rate). We are happy to include this additional result in the revision.
> > >
> > > 3) Our paper provides a new error metric for evaluating the quality of OT maps/kernels, along with efficient estimators that achieve provable guarantees under this metric. Domain adaptation is indeed a popular application of OT and EOT in machine learning, but our results are not restricted to any single application.
> > >
> > > [A]: Pooladian, Divol, Niles-Weed. "Minimax estimation of discontinuous optimal transport maps: The semi-discrete case". ICML 2023.
> > >
> > > [B]: Pooladian, Cuturi, Niles-Weed. "Debiaser Beware: Pitfalls of Centering Regularized Transport maps". ICML 2022.
> > >
> > > [C]: Deb, Ghosal, Sen. "Rates of Estimation of Optimal Transport Maps using Plug-in Estimators via Barycentric Projections". NeurIPS 2021.

---

> > > > ### Comment · Reviewer_LQyn · 2025-08-08
> > > >
> > > > I really appreciate your experiments, but I am sure that large-scale domain adaptation experiments would significantly improve the quality of your paper. One would like to see look at practical contribution of your work.
> > > >
> > > > I appreciate your clarifications about Sub-Gaussian and about error metrics, but I am really needed in large-scale experiments.

---

> > > > > ### Author Response · Authors · 2025-08-09
> > > > >
> > > > > Thank you for your continued discussion. If the reviewing team is in agreement that this addition is needed, we are happy to apply our previously proposed neural estimator to a large-scale domain adaptation task, using the image data sets suggested. However, we believe that our updated set of experiments exceeds the bar applied to OT map estimation papers published at top-tier ML venues in past years. Achieving strong performance for domain adaptation with image data often requires extensive hyperparameter and architecture searches, which we view as orthogonal to our main contribution (and worry would require discussion beyond the remaining page limit, if presented in the main body).

---

### Official Review · Reviewer_Q4zc · 2025-07-02

**Clarity:** 3
**Significance:** 2
**Originality:** 3
**Rating:** 4
**Confidence:** 2

**Summary:**

This paper introduces a refined $p$-Wasserstein-type error functional that incorporates an additional transportation kernel. Various forms of stability for marginal distributions with respect to the $p$-Wasserstein distance are established. Furthermore, the author studies the error when the reference transportation is close to the optimal one, under certain regularity assumptions. A synthetic experiment with a discontinuous optimal transport map is performed to test the accuracy of this error functional.

**Questions:**

In Section 4, line 243, the authors assume Hölder continuity of the optimal kernel for the $W_p$ problem. This seems quite limiting, as it may not hold in practice. Could the authors provide justification for this assumption?

**Ethical Concerns:**

["NO or VERY MINOR ethics concerns only"]

**Final Justification:**

The paper is well written and makes strong theoretical contributions. The authors provided clear and satisfactory responses to reviewers' concerns. It seems that there will be supporting experiments coming up; therefore, I recommend a weak acceptance at the moment.

**Limitations:**

Yes

**Quality:**

3

**Strengths And Weaknesses:**

Strengths:
This article is mathematically sophisticated and sound, with a well-organized presentation. The refined error functional sheds new light on the errors in approximating optimal transport, providing insights that were not captured in previous results.

Weakness:
Even though the new error functional $\mathcal{E}_p(\kappa; \mu, \nu)$ can be used to analyze the error near optimal transport maps (for instance, for the minimizer in the entropic estimator in Section 3), it does not appear to provide an algorithm for obtaining better approximations or a more scalable method for large data sets.

---

> ### Author Rebuttal · Authors · 2025-07-31
>
> We thank the reviewer for their thoughtful feedback and clarifying questions.
>
> Weakness: The reviewer fairly points out that we do not use our methods to design more accurate / scalable algorithms for truly large scale settings. As a minor point, we note that our framework opens the door for analysis of future algorithms under far weaker assumptions than currently standard. But, more importantly, we are optimistic that our techniques may have immediate applications for neural map estimation with large-scale data, via a connection to the Monge gap discussed below.
>
> As noted by Reviewer RJR5, there is a promising approach for neural map estimation using the so-called “Monge gap” [A], which we will discuss in the updated related work. When specialized to $p=1$, this work aims to learn a parameterized map $T$ minimizing the following:
> $$\\mathcal{E}’\_1(T;P,Q) := \underbrace{\mathbb{E}\_P[\\|T(X) - X\\|] - W\_1(P,T\_\sharp P)}\_{\text{Monge gap}} + \Delta(T\_\sharp P,Q),$$
> where $\Delta$ is some statistical divergence. In practice, they substitute P and Q with their empirical versions and use entropic OT (EOT) distances to approximate both $W\_1$ and $\Delta$, achieving impressive empirical results. We emphasize that these approximations preclude meaningful statistical guarantees without accounting for implicit bias or other regularization, since an optimal map for the empirical problem can perform arbitrarily bad on the population problem.
>
> We can show that, when $\Delta = W\_1$, the Monge gap objective $\mathcal{E}’\_1$ and our $\mathcal{E}\_1$ coincide up to a constant multiplicative factor of 2. Qualitatively similar results hold for $p>1$, though there is no longer a precise equivalence. Thus, the approach in [A] is quite close to performing empirical risk minimization with our transportation error (up to the use of EOT that is necessary for large-scale data sets). We will include this derivation in a new remark for Section 2. In some sense, this shows that our approach already lends itself to map fitting in practice. Moreover, if we were to directly plug in our $\mathcal{E}\_p$ objective into the neural map estimation experiments from [A] (substituting OT with EOT for tractability), we have the benefit of only having to perform back propagation through one EOT problem instead of two. We are optimistic that this will lead to a performance boost in practice. In the updated manuscript, we will supplement our synthetic experiments, which validated our theory, with a large-scale experiment that compares our performance to the Monge gap approach when learning maps for single-cell genomic data (mirroring the setup of Section 6.4 in [A]). Zooming out, we feel that our framework provides a solid statistical foundation for the practical approach to map estimation established in [A], and we hope that our theoretical insights will boost further practical developments.
>
> Second, our rounding estimation approach can be applied in a black-box way on top of any existing map estimator, including a neural estimator. In the added experiments with single-cell genomic data, we will take the resulting neural estimators and see how their test performance varies if we add on this rounding step, for varied choices of rounding partition (namely, tuning the size of the partition cells). It will be interesting to see if there is extra performance to be eked out of existing methods via the regularizing effect of rounding.
>
> **Question, Hölder continuity:** This is indeed a strong assumption, but it is in fact weaker than nearly all of the existing work on statistical rates for $L^2$ map estimation (with the exception of the non-comparable setup in [B] that requires $T^\star$ to be piecewise-constant). We agree that this is of less practical interest, which is why we focused on the unrestricted setting first, but we wanted to include this setting since it covered a large set of related work. There are some theoretical results when $p=2$ that guarantee the existence of a regular Brenier map (Caffarelli’s regularity theory - see, e.g., Theorem 3 of [C]), but these are admittedly not quantitative.
>
> We also note that our approach to robustifying estimation to Wasserstein perturbations (for Section 5) uses a black-box method for turning any kernel into a Hölder continuous kernel via Gaussian convolution (at the cost of introducing some bias). Thus, the continuity condition can be of practical interest even if it does not hold exactly in practice.
>
> [A]: Uscidda & Cuturi. “The Monge Gap: A Regularizer to Learn All Transport Maps”. ICML 2023.
> [B]: A.-A. Pooladian, V. Divol, and J. Niles-Weed. Minimax estimation of discontinuous optimal transport maps: The semi-discrete case. ICML 2023.
> [C]: Manole, Balakrishnan, Niles-Weed, & Wasserman. Plugin estimation of smooth optimal transport maps. The Annals of Statistics, 2024.

---

> > ### Comment · Reviewer_Q4zc · 2025-08-06
> >
> > The authors have addressed my concerns regarding the Hölder continuity assumption. I appreciate their effort in updating the synthetic experiments based on [A]. I plan to raise my score following these experimental updates.

---

### Official Review · Reviewer_YnBD · 2025-07-08

**Clarity:** 4
**Significance:** 3
**Originality:** 3
**Rating:** 5
**Confidence:** 2

**Summary:**

This paper develops a new theoretical framework for estimating _stochastic_ optimal transport (OT) maps. It broadens the classical deterministic Monge setting where OT maps existence is due to Brenier’s theorem, and it encompass scenarios where no unique deterministic map exists. Their framework relies on breaking down the estimation error into an _optimality gap_ (the excess transport cost over the true OT value) and a _feasibility gap_ (the Wasserstein distance between the map’s push-forward and the true target).

The analysis consists of breaking the overall error into two parts:

- The optimality gap: This is characterized by studying the **entropic kernel estimator**, which is used to study the optimality gap, leveraging entropic OT with a bias–variance trade-off and its sample complexity has been studied by Genevay et al. 2019 and Peyre et al. 2013.
- The feasibility gap: This is characterized by the **rounding estimator**, which discretizes the source, trims the target, solves a near-optimal discrete OT problem, then “rounds” back to the continuous domain. The sample complexity of this is analyzed Hölder continuous kernels.

**Questions:**

**Questions to the Authors**

I think the work tackles an important problem of studying finite-sample bounds for stochastic OT problems. I have the following question:
- As an OT practitioner, I have the following question, could the finite-sample rates inform how OT maps are parametrized and fitted?   Could you characterize the error functional (i.e. which error term dominates when) at different sample ranges?
- Does this sample complexity bounds hold for conditional OT maps where there is a family of OT maps is fitted instead of a single one? If not, could the authors provide intuition on how it may generalize to conditional OT problems?

**Ethical Concerns:**

["NO or VERY MINOR ethics concerns only"]

**Final Justification:**

The authors have provided satisfactory answers to my questions. Note that I am not an OT theory expert, please consider my opinion as a view point of an OT practitioner.

**Limitations:**

Not relevant

**Quality:**

4

**Strengths And Weaknesses:**

**Strengths**

- The paper is very well written.
- Studies an important problem of characterizing OT maps when Brenier theorem does not hold, which is a common occurrence in real-world, finite-sample settings.
- The proposed error functional is general, and it applies when no deterministic OT map exists, covering fully stochastic real-world settings common in domain adaptation and single-cell genomics.
- Finite-sample upper bounds match minimax lower bounds under the assumption of sub-Gaussianity (or finite moments) without requiring density smoothness.
- The stability lemmas for the error functional enable decouple the local (Wasserstein) and global (TV) robustness, yielding optimal rates under strong corruption models.
- The rates are corroborated empirically. Both entropic and rounding estimators are implemented via existing entropic OT solvers or discrete OT routines.


**Weaknesses**
- None that I can point to.
- Given my limited expertise in theory of OT, I cannot judge the novelty of the theoretical claims made in this work. I hope that the other reviewers and the AC could evaluate these aspects.

---

> ### Author Rebuttal · Authors · 2025-07-31
>
> We thank the reviewer for their thoughtful feedback and clarifying questions.
>
> **Summary clarification:** We wanted to point out that bounding both gaps is required in the analysis of both the entropic kernel estimator and the rounding estimator. If we were to distinguish the estimators’ analysis, we would say that the entropic kernel estimator relies on stability of $\\mathcal{E}\_p$ under Wasserstein perturbations, while the rounding estimator relies on TV stability. We will mention this distinction at the start of Section 3. Also, the analysis of the rounding estimator does not require Hölder continuous kernels (that case is treated by a separate DRO estimator introduced in Section 4).
>
> **Q1a “could the finite-sample rates inform how OT maps are parametrized and fitted?”:**
> We agree that using this work to inform map estimation in practice is a very natural direction. As noted by Reviewer RJR5, there is a promising approach for neural map estimation using the so-called “Monge gap” [A], which we will discuss in the updated related work. When specialized to $p=1$, this work aims to learn a parameterized map $T$ minimizing the following:
> $$\mathcal{E}’\_1(T;P,Q) := \underbrace{\mathbb{E}\_P[\\|T(X) - X\\|] - W\_1(P,T\_\sharp P)}\_{\text{Monge gap}} + \Delta(T\_\sharp P,Q),$$
> where $\Delta$ is some statistical divergence. In practice, they substitute $P$ and $Q$ with their empirical versions and use entropic OT (EOT) distances to approximate both $W\_1$ and $\Delta$, achieving impressive empirical results. We emphasize that these approximations preclude meaningful statistical guarantees without accounting for implicit bias or other regularization, since an optimal map for the empirical problem can perform arbitrarily bad on the population problem.
>
> We can show that, when $\Delta = W\_1$, the Monge gap objective $\mathcal{E}’\_1$ and our $\mathcal{E}\_1$ coincide up to a constant multiplicative factor of 2. Qualitatively similar results hold for $p>1$, though there is no longer a precise equivalence. Thus, the approach in [A] is quite close to performing empirical risk minimization with our transportation error (up to the use of EOT that is necessary for large-scale data sets). We will include this derivation in a new remark for Section 2. In some sense, this shows that our approach already lends itself to map fitting in practice. Moreover, if we were to directly plug in our $\mathcal{E}\_p$ objective into the neural map estimation experiments from [A] (substituting OT with EOT for tractability), we have the benefit of only having to perform back propagation through one EOT problem instead of two. We are optimistic that this will lead to a performance boost in practice. In the updated manuscript, we will supplement our synthetic experiments, which validated our theory, with a large-scale experiment that compares our performance to the Monge gap approach when learning maps for single-cell genomic data (mirroring the setup of Section 6.4 in [A]). Zooming out, we feel that our framework provides a solid statistical foundation for the practical approach to map estimation established in [A], and we hope that our theoretical insights will boost further practical developments.
>
> Second, our rounding estimation approach can be applied in a black-box way on top of any existing map estimator, including a neural estimator. In the added experiments with single-cell genomic data, we will take the resulting neural estimators and see how their test performance varies if we add on this rounding step, for varied choices of rounding partition (namely, tuning the size of the partition cells). It will be interesting to see if there is extra performance to be eked out of existing methods via the regularizing effect of rounding.
>
> Third, we call the reviewer’s attention to our Remark 4, which draws parallels between our DRO procedure and existing Lipschitz regularization techniques.
>
> Finally, we remark that our current synthetic experiments were computationally bottlenecked by high-precision OT computations, which were necessary to validate our statistical guarantees. The Monge gap work sidesteps these by using EOT with a fixed precision, trading some accuracy for scalability. Another regularization approach which may be worth future exploration is that of sliced OT, where one works only with low dimensional projections of the data distributions. We will add a remark to the end of Section 2 on extensions of our framework to regularized OT problems, which are key for large-scale applications.
>
> **Q1b “Could you characterize the error functional (i.e. which error term dominates when) at different sample ranges?”:**
> This is an interesting question to explore. The answer would appear to depend on quite a few factors, including the kernel estimator, the sample size, the source distribution, and the target distribution. We are unaware of any simple primitive conditions that would help distinguish the case. To investigate this, we will update all of our experiments to separate the contributions of the two error terms and include some discussion on the resulting balance. For our existing synthetic experiments, we note that the feasibility gap was typically larger than the optimality gap, but we will also include this fine-grained data for the neural estimation experiments promised above.
>
> **Q2, extension to conditional OT:**
> This is a great question, certainly of relevance in important applications. In general, we believe that our framework is modular enough to apply to a wide-range of OT variants, including conditional OT. For this case, one could extend existing conditional OT works to measure performance with an $\mathcal{E}\_2$-type functional instead of $L^2$. Depending on the modeling assumptions, one could statistically investigate questions of finite-sample risk. We expect meaningful bounds to require additional regularity assumptions on the relationship between contexts and maps/kernels. We view this as a promising direction for future research and will add a discussion to the summary of our final version.
>
> [A]: Uscidda & Cuturi. “The Monge Gap: A Regularizer to Learn All Transport Maps”. ICML 2023.

---

### Decision · Program_Chairs · 2025-09-17

**Decision:**

Accept (poster)

**Comment:**

The work studies consistent estimators for stochastic transport maps. This is an interesting and novel problem as the existing literature is focused on estimation of (deterministic) Monge maps. The idea is to define a so-called transportation error, which quantifies the optimality of any given stochastic transport map (kernel). It is simply a sum of a feasibility and an optimality term. Firstly, it is shown that the entropic OT based kernel is consistent as per this transportation error. A new rounding esitmator is also presented along with its consistency in theorem2. This new estimator turns out to be optimal nearly matching the lower bounds. One key feature is no regulairty assumptions are made. Under the (regulairty) assumption of holder conts kernels, a DRO based estimator is also analyzed.

The reviewers also broadly agree that all these results are interesting and will be useful for the ML community. The paper is also well written with a clear flow and organization.

As promised in the rebuttal, it will help if the related work with monge gap and the additional simulations are included in the final version.